# YTHDF2 facilitates aggresome formation via UPF1 in an m⁶A-independent manner

Hyun Jung Hwang[1,2], Tae Lim Park [1,2], Hyeong-In Kim [1,2], Yeonkyoung Park[1], Geunhee Kim [1], Chiyeol Song[1], Won-Ki Cho [1]✉ & Yoon Ki Kim [1]✉

YTHDF2 has been extensively studied and typified as an RNA-binding protein that specifically recognizes and destabilizes RNAs harboring $N^6$-methylade-nosine (m⁶A), the most prevalent internal modification found in eukaryotic RNAs. In this study, we unravel the m⁶A-independent role of YTHDF2 in the formation of an aggresome, where cytoplasmic protein aggregates are selectively sequestered upon failure of protein homeostasis mediated by the ubiquitin-proteasome system. Downregulation of YTHDF2 in HeLa cells reduces the circularity of aggresomes and the rate of movement of misfolded polypeptides, inhibits aggresome formation, and thereby promotes cellular apoptosis. Mechanistically, YTHDF2 is recruited to a misfolded polypeptide-associated complex composed of UPF1, CTIF, eEF1A1, and DCTN1 through its interaction with UPF1. Subsequently, YTHDF2 increases the interaction between the dynein motor protein and the misfolded polypeptide-associated complex, facilitating the diffusion dynamics of the movement of misfolded polypeptides toward aggresomes. Therefore, our data reveal that YTHDF2 is a cellular factor involved in protein quality control.

Not all polypeptides are always folded properly in nature. Misfolded polypeptides arise within cells due to many intrinsic and extrinsic factors, such as amino acid substitutions caused by genetic mutations or transcriptional errors, premature translation termination (which generates defective ribosomal products), aberrant post-translational modifications, and a variety of environmental stresses[1]. These misfolded polypeptides are subjected to multi-layered protein quality control (PQC) pathways that help cells maintain protein homeostasis and mitigate proteotoxic stresses[2–7]. Improper PQC causes the accumulation of misfolded polypeptides and is associated with numerous human diseases, including neurodegenerative diseases[8–10].

As the first safeguard for protein homeostasis, newly generated misfolded polypeptides are rapidly refolded into their native conformation by the action of molecular chaperones. When this process is overwhelmed or compromised, the ubiquitin-proteasome system (UPS) covalently labels the misfolded polypeptides with ubiquitin and rapidly degrades these labeled proteins. However, many misfolded polypeptides escape the UPS and form small cytoplasmic aggregates

that eventually form a single large cytoplasmic inclusion called the "aggresome"[11–13]. Notably, aggresomes containing misfolded polypeptides are biochemically and morphologically similar to Lewy bodies (spherical cytoplasmic inclusions containing α-synuclein aggregates) observed in Parkinson's disease and Lewy body dementia[8–10], thus highlighting the physiological importance of aggresomes.

Aggresome formation and clearance occur through a series of sequential steps in the cytoplasm. Misfolded polypeptides or their small aggregates are selectively recognized by three major aggresome-targeting cellular machineries: (i) histone deacetylase 6 (HDAC6); (ii) a complex consisting of BCL2-associated athanogene 3 (BAG3) and its associated chaperones and co-chaperones; and (iii) the CED complex composed of nuclear cap-binding complex (CBC)-dependent translation initiation factor (CTIF), eukaryotic translation elongation factor 1 alpha 1 (eEF1A1), and dynactin 1 (DCTN1)[14]. The molecular features of misfolded polypeptides and the cellular environment that determine the preference for three aggresome-targeting machineries remain an open question. After selective recognition, the small cytoplasmic

[1]Department of Biological Sciences, Korea Advanced Institute of Science and Technology, Daejeon 34141, Republic of Korea. [2]These authors contributed equally: Hyun Jung Hwang, Tae Lim Park, Hyeong-In Kim. ✉e-mail: wonkicho@kaist.ac.kr; yk-kim@kaist.ac.kr

aggregates complexed with aggresome-targeting machineries associate with dynein motor proteins, move toward the minus end of microtubules, and accumulate at the periphery of the nucleus, forming an aggresome[11–13]. Finally, the misfolded polypeptides accumulated within the aggresomes are removed by aggrephagy, a selective autophagy pathway[11–13].

The CED complex-mediated targeting of misfolded polypeptides toward aggresomes is guided by upstream frameshift 1 (UPF1), a key factor in the best-characterized mRNA surveillance mechanism—non-sense-mediated mRNA decay (NMD), by which aberrant mRNAs harboring a premature termination codon are selectively downregulated[15]. Single-particle tracking using line-scan confocal microscopy in live cells showed that UPF1 increases the frequency and fidelity of movement of CTIF particles (possibly in association with misfolded polypeptides) toward aggresomes[16]. Notably, other NMD factors, such as SMG5, SMG6, SMG7, and PNRC2, have no significant effect on aggresome formation, indicating a specific role of UPF1 in both protein and mRNA surveillance pathways. In support of this notion, arginine-rich dipeptide repeats generated from the repeat sequences of GGGGCC in the C9ORF72 gene inhibit NMD efficiency and induce dipeptide repeat toxicity in neuronal cells, which is reversed by UPF1 overexpression, suggesting the protective role of UPF1 in neurons[17]. UPF1 can also function as a neuroprotective agent by reducing dipeptide repeat toxicity without affecting NMD efficiency[18].

UPF1 is a multi-talented protein involved in NMD and many other mRNA decay pathways[15]. Recently, it was also reported that UPF1 triggers the rapid degradation of RNAs containing $N^6$-methyladenosine (m⁶A; modified adenosine with a methyl group at the 6th position of the amino group) through its direct interaction with an m⁶A-reader protein, YT521-B homology domain-containing family protein 2 (YTHDF2)[19,20]. The m⁶A modification is the most abundant internal mRNA modification. It takes place in the nucleus with the action of the m⁶A writer complex composed of methyltransferase-like protein 3 (METTL3), METTL14, and other auxiliary factors[21–26]. Notably, m⁶A can be reversed to adenosine by m⁶A eraser proteins: α-ketoglutarate-dependent dioxygenase AlkB homolog 5 protein (ALKBH5) and fat mass and obesity-associated protein (FTO)[23–25]. The installed m⁶A is selectively recognized by m⁶A reader proteins, such as YTHDF1, YTHDF2, and YTHDF3, and consequently affects diverse molecular events[23–25,27–29]. As mentioned above, the UPF1-YTHDF2 interaction promotes decapping, followed by 5′–3′ degradation of YTHDF2-bound m⁶A mRNAs[19,20].

In this study, we unravel the m⁶A-independent role of YTHDF2 in aggresome formation, highlighting that YTHDF2 plays dual roles in regulating gene expression—as a cellular factor involved in PQC and as the well-characterized m⁶A-recognizing RNA-binding protein.

## Results

### The m⁶A reader protein, YTHDF2, is involved in aggresome formation

A recent study showed that YTHDF2 interacts with UPF1[19,20], which guides proper targeting of misfolded polypeptides to the aggresome[16]. This interaction led us to investigate the possible connection between m⁶A modification and aggresome formation. To this end, we first tested the possible effects of m⁶A writer, reader, and eraser on the formation of aggresomes containing misfolded polypeptides generated by puromycin (puro) treatment. The treatment of HeLa cells with puro triggers premature termination of elongating ribosomes, releasing truncated polypeptides conjugated with puro (polypeptidyl-puro)[30]. Polypeptidyl-puro corresponds to misfolded defective ribosomal products and is transported to aggresomes upon treatment with MG132, a potent proteasome inhibitor[16,30–32].

As expected, while most cells lacked aggresomes under normal conditions (Supplementary Fig. 1a), ~60% of cells exhibited distinct aggresomes containing polypeptidyl-puro under MG132-treated conditions (Fig. 1a, b). Notably, the percentage of cells exhibiting aggresomes was significantly reduced upon the downregulation of YTHDF2 or DCTN1 (a component of the CED complex) using small interfering RNA (siRNA) (Fig. 1a, b). In contrast, downregulation of the m⁶A writer (METTL3 or METTL14), m⁶A eraser (FTO), or SMG6 (a well-characterized NMD factor, which was shown to be not involved in aggresome formation and therefore served as a negative control[16]) only marginally affected the population of cells exhibiting aggresomes (Fig. 1a, b). These results suggest that YTHDF2 promotes the efficient formation of aggresomes of misfolded polypeptides, possibly in an m⁶A-independent manner. In agreement with the conclusion, downregulation of YTHDF2, but not of METTL3, METTL14, or FTO, reduced the population of cells exhibiting aggresomes containing another type of misfolded polypeptide—green fluorescent protein (GFP)-fused cystic fibrosis transmembrane conductance regulator ΔF508 (CFTR-ΔF508) (Supplementary Fig. 1b, c). This CFTR variant has a single amino acid deletion (phenylalanine at position 508) and is known to form aggresomes in cultured cells upon MG132 treatment[12,31,33]. Specific downregulation of each protein under our experimental conditions was confirmed using western blotting (Supplementary Fig. 1d).

Next, we investigated whether YTHDF2 itself is transported to aggresomes or it has aggresome-targeting features. YTHDF2 is composed of a intrinsically disordered low-complexity N-terminal half rich in proline, glutamine, and asparagine and a C-terminal half with a YTH domain responsible for m⁶A recognition[34,35]. Immunostaining revealed that FLAG-YTHDF2 and endogenous YTHDF2 were evenly distributed throughout the cytoplasm regardless of MG132 treatment (Fig. 1c and Supplementary Fig. 1e). In addition, FLAG-YTHDF-C lacking the N-terminal half was observed in both the nucleus and the cytoplasm, and its localization was not affected by MG132 treatment. In contrast, FLAG-YTHDF-N lacking the C-terminal half was hardly observed under normal conditions. Of note, we observed strong colocalization of FLAG-YTHDF-N and CFTR-ΔF508 after MG132 treatment. These data suggest that the N-terminal half of YTHDF2 has intrinsic features that can induce protein misfolding and aggresome formation. In support of our conclusion, recent studies have shown that a low-complexity region, such as the N-terminal half of YTHDF2, tends to drive liquid-like phase separation and consequently forms liquid droplets[36–39].

### YTHDF2 is selectively complexed with CED components

Three paralogs belonging to the YTHDF family have been well-characterized: YTHDF1, YTHDF2, and YTHDF3. Previous studies have suggested a distinct role for each paralog in RNA decay and/or translation. However, considering that all three paralogs have a high similarity of amino acid sequences, are commonly localized in the cytoplasm, and share their target transcripts[40–43], it is plausible that the three paralogs can function in a compensatory or redundant manner. In fact, a context-dependent compensatory role among YTHDF1–3 has been reported previously[44,45]. Furthermore, YTHDF proteins are known to interact with each other (Shi et al., 2017). Therefore, we investigated whether all three YTHDF paralogs are comparably or redundantly involved in aggresome formation.

We observed that the downregulation of YTHDF2, but not of YTHDF1 or YTHDF3, caused inefficient formation of aggresomes containing polypeptidyl-puro (Fig. 2a, b) and CFTR-ΔF508 (Fig. 2c, d) upon MG132 treatment. Specific depletion of endogenous YTHDF protein by siRNA treatment was validated using western blotting (Fig. 2e). We also tested the intracellular distribution of three YTHDF paralogs based on previous reports showing the enrichment of YTHDF paralogs in stress granules (SGs)—the cytoplasmic non-membranous compartment where translationally repressed messenger ribonucleoproteins (mRNPs) are enriched under stress conditions. Treatment with sodium arsenite for 1 h resulted in the efficient formation of SGs containing G3BP1 (a SG marker protein; Supplementary Fig. 2a). Notably, all YTHDF paralogs co-localized with SGs containing G3BP1, with

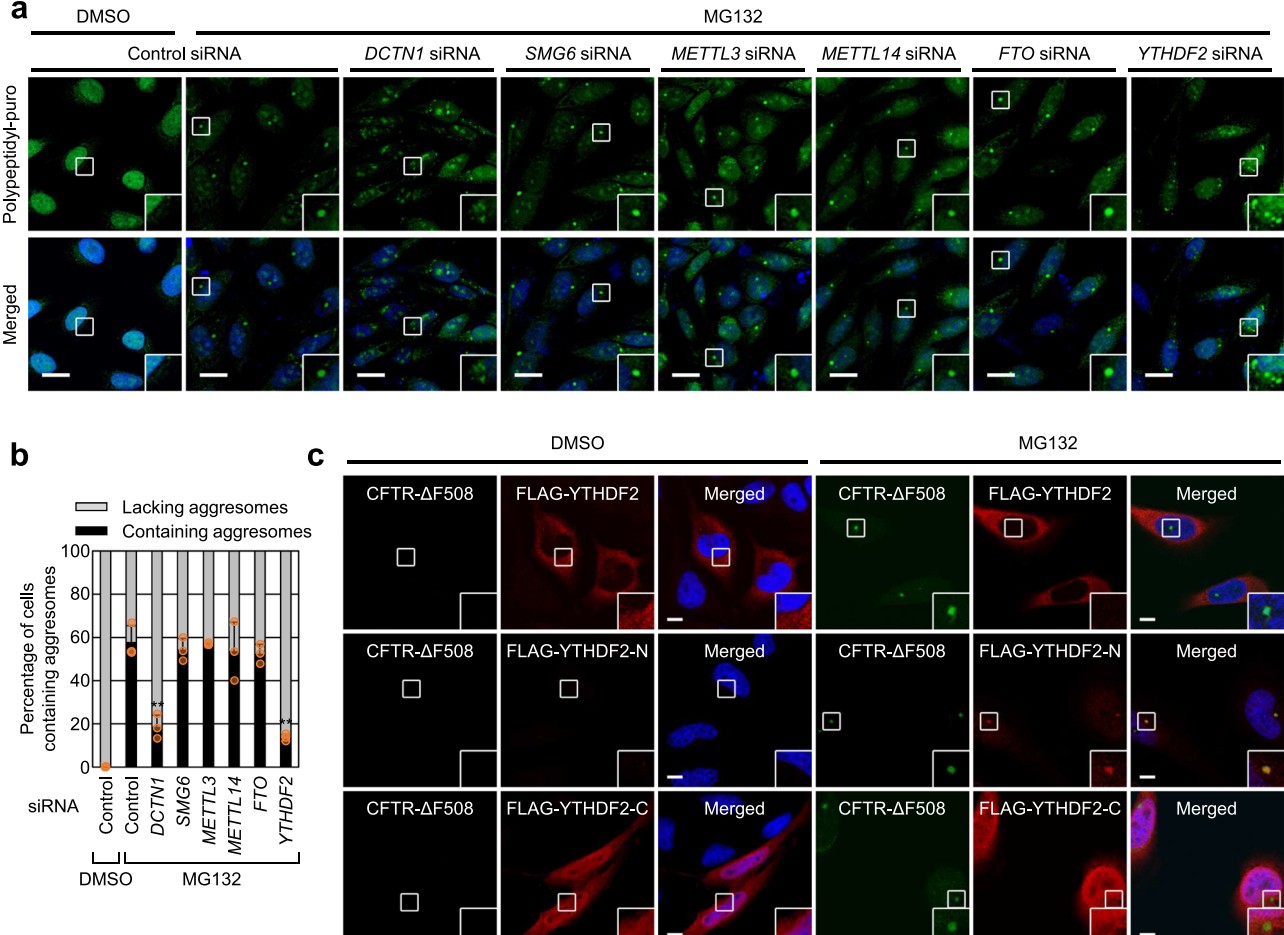

**Fig. 1 | YTHDF2 promotes aggresome formation in an m⁶A-independent manner. a** Immunostaining of polypeptidyl-puro aggresome. HeLa cells were transfected using the indicated siRNA. Two days later, the cells were treated with either DMSO or MG132 for 12 h and with puromycin for 1 h before fixation. Nuclei were visualized using DAPI (blue). Enlarged images of the boxed regions are shown in the lower-right corner of each image. All data represent at least three biologically independent replicates ($n = 3$). Scale bar, 10 μm. **b** The percentage of the cells exhibiting aggresomes containing polypeptidyl-puro in **a**. More than 50 cells from each experiment were quantitated. Two-tailed, equal-sample variance Student's $t$ test was carried out to calculate the $P$ values. **$P < 0.01$ (The exact $P$ values are provided in Source Data file). **c** Immunostaining of CFTR-ΔF508 (green) and FLAG-YTHDF2, -YTHDF2-N, or -YTHDF2-C (red). HeLa cells stably expressing CFTR-ΔF508 were transiently transfected with the indicated plasmid expressing FLAG-YTHDF2, -YTHDF2-N, or -YTHDF2-C. The cells were treated with either DMSO or MG132 for 12 h before fixation. Nuclei were visualized using DAPI (blue). Scale bar, 10 μm; $n = 3$.

comparable levels under our experimental conditions. These data indicate that while all three YTHDF paralogs have the ability to be located in SGs with comparable efficiency, only YTHDF2 specifically contributes to aggresome formation. We also observed that while treatment with nocodazole (a potent chemical that inhibits microtubule polymerization) drastically disrupted microtubules, YTHDF2 downregulation only marginally affected global microtubule formation and distribution (Supplementary Fig. 2b). Therefore, it is unlikely that YTHDF2 affects aggresome formation by altering the intracellular distribution of microtubules.

Next, we conducted immunoprecipitation (IP) experiments using HEK293T cell extracts and a specific antibody against each YTHDF paralog. CTIF, eEF1A1, DCTN1 (all of which are components of the CED complex), and UPF1 (which associates with the CED complex) were enriched in the IP of YTHDF2, but not of YTHDF1 or YTHDF3 (Fig. 2f), indicating a specific association between YTHDF2 and the CED complex.

### YTHDF2 associates with the CED complex and misfolded polypeptides via the UPF1

Considering that (i) YTHDF2 selectively associates with the CED complex (Fig. 2f), (ii) YTHDF2 binds to UPF1 (Fig. 2f)[19,26], and (iii) UPF1

interacts with the N-terminal region of CTIF and helps the CED complex maintain its integrity[16], it is plausible that YTHDF2 may interact with the CED complex by binding to UPF1[16]. This possibility was tested using a series of IP experiments as follows.

First, UPF1, eEF1A1, DCTN1, and YTHDF2 coimmunoprecipitated with FLAG-CTIF in a way that was independent of MG132 treatment (Supplementary Fig. 3a). Notably, comparable amounts of UPF1, eEF1A1, and DCTN1 were enriched in the IPs of CTIF even after the cells were depleted of YTHDF2 (Supplementary Fig. 3a), indicating that YTHDF2 is not actively involved in the maintenance of CED integrity. Second, the UPF1 and CED components coimmunoprecipitated with endogenous YTHDF2 independently of MG132 treatment (Fig. 3a). Notably, the amounts of coimmunoprecipitated CED components were significantly reduced by UPF1 downregulation, suggesting that UPF1 functions as a protein linker between YTHDF2 and the CED complex. Third, IP experiments using Myc-YTHDF2-wild type (WT), Myc-YTHDF2-Δ101-168, or Myc-YTHDF2-Δ101-200 showed that UPF1 and all CED components were enriched in the IP of Myc-YTHDF2-WT, but not in those of Myc-YTHDF2-Δ101-168 and Myc-YTHDF2-Δ101-200 (Fig. 3b), both of which are known to lack the UPF1-binding region[19]. Fourth, on the basis that UPF1 hyperphosphorylation promotes its association with the CED complex in an MG132 treatment-independent

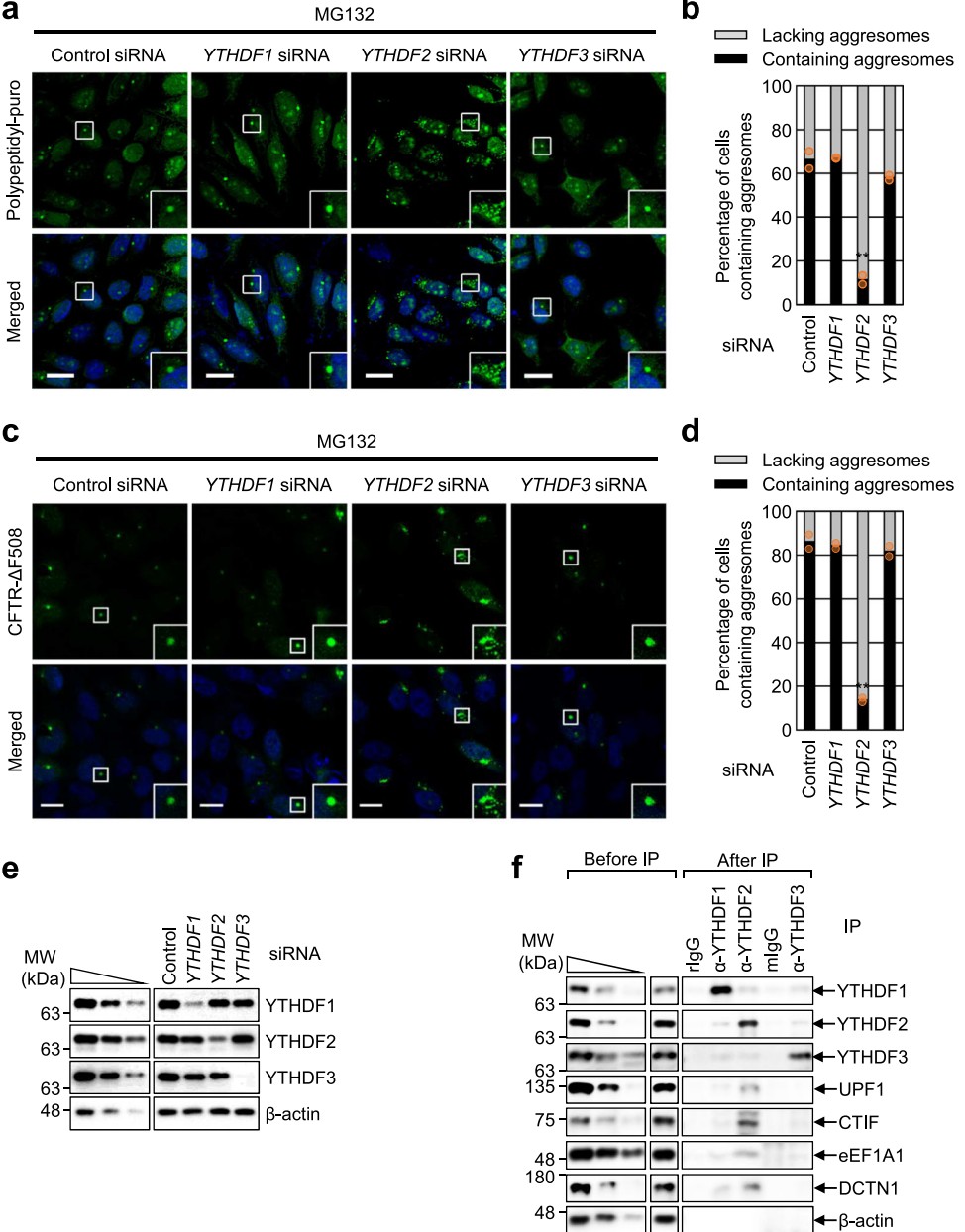

**Fig. 2 | YTHDF2 is the m⁶A reader protein that is specifically involved in aggresome formation. a** Immunostaining of aggresomes containing polypeptidyl-puro. HeLa cells were transfected with *YTHDF1*, *YTHDF2*, or *YTHDF3* siRNA. Two days later, the cells were treated with MG132 for 12 h and puromycin for 1 h before fixation. Scale bar, 10 µm; $n = 2$. **b** The percentage of the cells either containing or lacking aggresomes of polypeptidyl-puro after siRNA treatment in **a**. For quantitating the relative percentage of cells containing aggresomes, more than 50 cells from each experiment were examined. Two-tailed, equal-sample variance Student's *t* test was carried out to calculate the *P* values; **$P < 0.01$ (The exact *P* values are provided in Source Data file). **c** Immunostaining of CFTR-ΔF508 aggresome. HeLa

cells stably expressing CFTR-ΔF508 were transfected with *YTHDF1*, *YTHDF2*, or *YTHDF3* siRNA. The cells were treated with MG132 for 12 h before fixation. Scale bar, 10 µm; $n = 2$. **d** The percentage of the cells either containing or lacking aggresomes of CFTR-ΔF508 in **c**. **$P < 0.01$. **e** Confirmation of specific downregulation of the tested proteins in **a**–**d**. The total-cell lysate was serially diluted three-fold and loaded in the three leftmost lanes to demonstrate that western blotting is semi-quantitative in our experimental conditions. **f** Immunoprecipitation (IP) of endogenous YTHDF1, YTHDF2, or YTHDF3. HEK293T cells were treated with MG132 for 12 h before harvesting. IP experiments were performed with the indicated antibody, rabbit IgG (rIgG), or mouse IgG (mIgG) using extracts of HEK293T cells.

manner[16], we carried out IP experiments using UPF1-WT or its variants: hyperphosphorylated UPF1 (UPF1-HP) and UPF1-HP-12A. UPF1-HP, which harbors two amino acid substitutions (G495R and G497E), is hyperphosphorylated due to the lack of helicase and ATPase activities[16,46–49]. UPF1-HP-12A harbors additional 12 amino acid substitutions (at positions experimentally validated to be phosphorylated by SMG1, a well-characterized UPF1 kinase). Because of these additional substitutions, UPF1-HP-12A was shown to be inefficiently phosphorylated, although it contained G495R and G497E substitutions[16,46].

The IP results showed that a greater amount of YTHDF2 and CED components coimmunoprecipitated with UPF1-HP relative to UPF1-WT and UPF1-HP-12A, regardless of MG132 treatment (Supplementary Fig. 3b). Lastly, the CED complex is known to associate with misfolded polypeptides with the help of the chaperone activity of eEF1A1[50–52]. Of note, this association is abolished when cells are depleted of UPF1[16,31,53]. Based on these observations, we investigated the possible role of YTHDF2 in the recognition of misfolded polypeptides by the CED complex. The results of the IP experiments using an antibody against

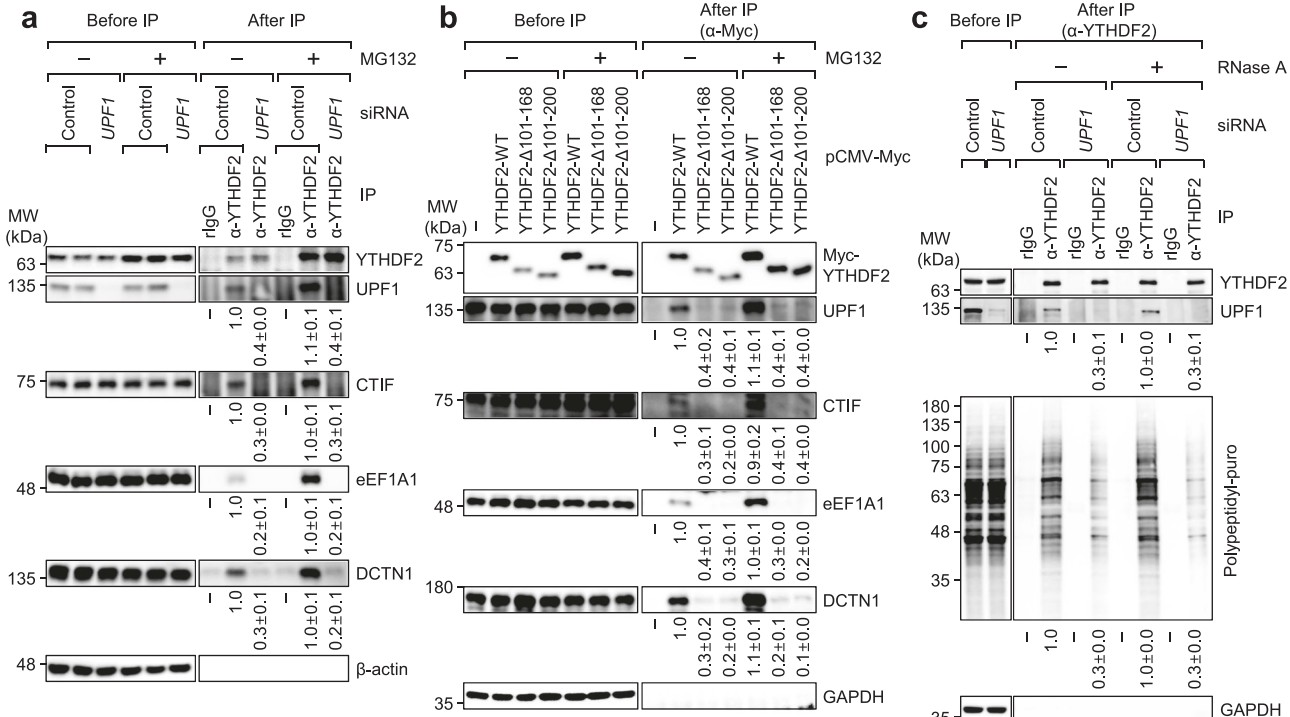

**Fig. 3 | UPF1 links YTHDF2 and the CED-misfolded polypeptide complex. a** IPs of endogenous YTHDF2 using extracts of HEK293T cells either depleted or not depleted of UPF1. The cells were transfected with *UPF1* siRNA. Two days later, the cells were treated with either DMSO or MG132 for 12 h before harvesting. Protein samples before or after IPs were analyzed by western blotting. The signal intensities of western blots after IPs were quantitated using ImageJ. The values of coimmunoprecipitated proteins were normalized to those of immunoprecipitated YTHDF2. The values obtained from YTHDF2 IPs in undepleted cells were arbitrarily set to 1.0; *n* = 3. **b** IPs of Myc-YTHDF2-WT or its variants using extracts of HEK293T cells. The

cells were transfected with Myc-YTHDF2-WT or its variant and treated with either DMSO or MG132 for 12 h before harvesting. Relative band intensities of proteins in the IPs of Myc-YTHDF2-WT were arbitrarily set to 1.0; *n* = 3. **c** IPs of endogenous YTHDF2 using extracts of cells. HEK293T cells were transfected with *UPF1* siRNA. Two days later, the cells were treated with MG132 for 12 h and puromycin for 1 h before harvesting. Total-cell extracts were either treated or not treated with RNase A before IP. The cell lysate before IP was serially diluted three-fold and loaded in the three leftmost lanes. Relative band intensities of proteins after IP of YTHDF2 in undepleted cells were arbitrarily set to 1.0; *n* = 3.

endogenous YTHDF2 revealed that polypeptidyl-puro was efficiently enriched in the IP of endogenous YTHDF (Fig. 3c). The observed enrichment was not affected by RNase A treatment but significantly reduced after UPF1 downregulation (Fig. 3c and Supplementary Fig. 3c), indicating that YTHDF2 associates with the misfolded polypeptides in a UPF1-dependent manner and an RNA-independent manner. Collectively, our IP data provide compelling evidence that UPF1 bridges the interaction between YTHDF2 and the CED complex associated with misfolded polypeptides.

## YTHDF2–UPF1 interaction efficiently promotes aggresome formation

We next investigated whether the YTHDF2–UPF1 interaction contributes to the formation of an aggresome containing misfolded polypeptides. To this end, we carried out complementation experiments using siRNA-resistant Myc-YTHDF2 (YTHDF2[R])-WT, -R527A, -Δ101–200, -Δ101–200/R527A, -Δ101–168, or -Δ169–200 (Supplementary Fig. 4a). Myc-YTHDF[R]-R527A contains a single amino acid substitution at position 527 within the YTH domain and, consequently, lacks RNA-binding ability[35]. Myc-YTHDF2[R]-Δ101–200 and -Δ101–168 lack the UPF1-interacting region[19]. Myc-YTHDF2[R]-Δ101–200/R527A lacks both the UPF1-binding region and RNA-binding ability. Myc-YTHDF2[R]-Δ169–200 lacks a binding site for CNOT1 (a component of the deadenylase complex) but contains a UPF1-binding region[19]. The specific interaction of each YTHDF2[R]-WT or its variant with endogenous UPF1 was confirmed using IP experiments (Fig. 4a).

The population of HeLa cells exhibiting distinct aggresomes containing the stably expressed CFTR-ΔF508 upon treatment with

MG132 was reduced from approximately 75% to 25%, when the cells were depleted of endogenous YTHDF2 (Fig. 4b, c). Under these conditions, the population of HeLa cells exhibiting the aggresome was significantly restored by the expression of Myc-YTHDF2[R]-WT, -R527A, and -Δ169-200, but not of Myc-YTHDF2[R]-Δ101-200, Δ101-200/R527A, and -Δ101-168 (Fig. 4b, c). Specific downregulation of endogenous YTHDF2 using siRNA and the expression of YTHDF2[R] variants to comparable levels were confirmed using western blotting (Supplementary Fig. 4b). It should be noted that a recent study reported that YTHDF2 has the ability to interact with m[1]A, another type of RNA modification[54]. However, we observed that cellular RNAs harboring the modified nucleotides m[6]A or $N^1$-methyladenosine (m[1]A) were not enriched in the aggresome containing Myc-UPF1-HP under MG132-treated conditions (Supplementary Fig. 4c). Altogether, our observations indicate that (i) a specific interaction between YTHDF2 and UPF1 is involved in the efficient formation of aggresomes, and (ii) the RNA-binding ability of YTHDF2 is not essential for aggresome formation.

## Impaired interaction between YTHDF2 and UPF1 causes cellular apoptosis

It is generally considered that aggresome formation contributes to cellular resistance to proteotoxic stresses. Inefficient aggresome formation and accumulation of misfolded polypeptides as small aggregates throughout the cytoplasm trigger proteotoxic stresses, inducing cellular apoptosis[16,31,33]. Therefore, we investigated the possible role of YTHDF2 in cellular response to proteotoxic stresses. To this end, we used a terminal deoxynucleotidyl transferase

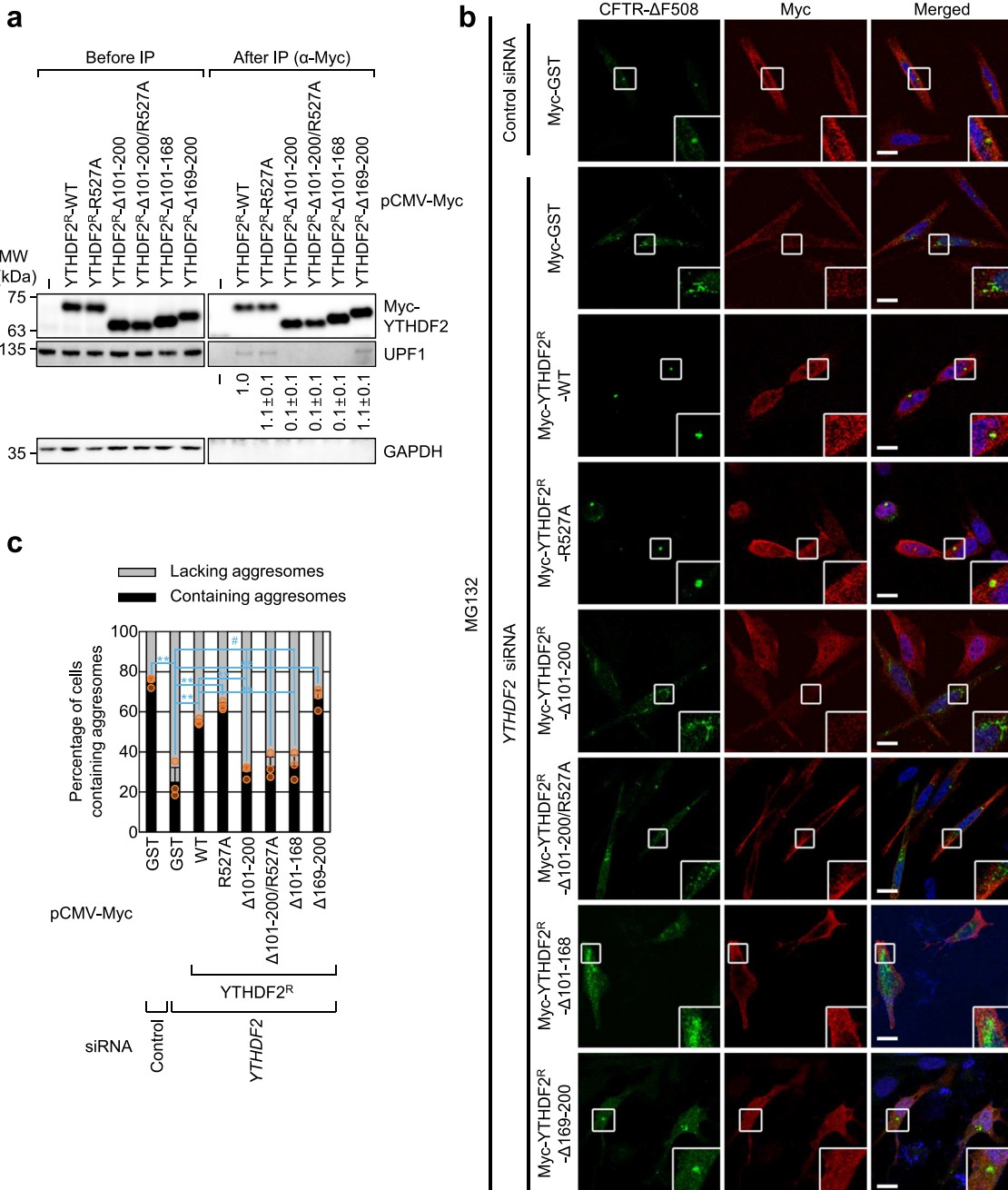

**Fig. 4 | Efficient aggresome formation depends on YTHDF2 binding to UPF1.**
**a** IPs using Myc-YTHDF2$^R$ (either WT or its variant). HEK293T cells were transiently transfected with a plasmid expressing Myc-YTHDF2$^R$ (either WT or its variant). Two days after transfection, the cell extracts were subjected to IPs using an α-Myc antibody. The relative level of co-immunopurified endogenous UPF1 in the IP of Myc-YTHDF2$^R$-WT was arbitrarily set to 1.0; $n$ = 3. **b** Immunostaining of CFTR-ΔF508 aggresome (green) and either Myc-GST or siRNA-resistant (R) Myc-YTHDF2$^R$ [either WT or one of the variants] (red). HeLa cells stably expressing CFTR-ΔF508 were transfected with either control or *YTHDF2* siRNA. One day later, the cells were retransfected with the indicated plasmid expressing either Myc-GST or Myc-

YTHDF2$^R$ (either WT or one of the variants). The cells were treated with MG132 for 12 h before fixation. Scale bar, 10 μm; $n$ = 3. **c** The percentage of the cells either containing or lacking aggresomes of CFTR-ΔF508. To assess the effect of exogenously expressed Myc-YTHDF2$^R$ on aggresome formation, the distributions of CFTR-ΔF508 were determined only in the cells expressing either exogenous Myc-GST or Myc-YTHDF2$^R$ (either WT or its variants). For quantitating the relative percentage of cells containing aggresomes, >50 cells from three biological replicates were examined. Two-tailed, equal-sample variance Student's $t$ test was carried out to calculate the $P$ values; #, not significant; **$P$ < 0.01 (The exact $P$ values are provided in Source Data file).

dUTP nick end labeling (TUNEL) assay to monitor cells undergoing apoptosis (Fig. 5). When HeLa cells stably expressing CFTR-ΔF508 were treated with MG132, cells underwent apoptosis at a basal level. In contrast, when YTHDF2 was downregulated under the same conditions, approximately 60% of cells underwent apoptosis. Notably, the observed level of apoptosis was significantly reversed by the expression of Myc-YTHDF2$^R$-WT, -R527A, and -Δ169–200 (all

of which have the ability to interact with UPF1), but not of Myc-YTHDF2$^R$-Δ101–200, Δ101–200/R527A, and -Δ101–168 (all of which lack the UPF1-binding region). Together with our observation that the YTHDF2–UPF1 interaction contributes to the efficient formation of aggresomes (Fig. 4), these results indicate that YTHDF2 renders cells more resistant to proteotoxic stresses via its interaction with UPF1.

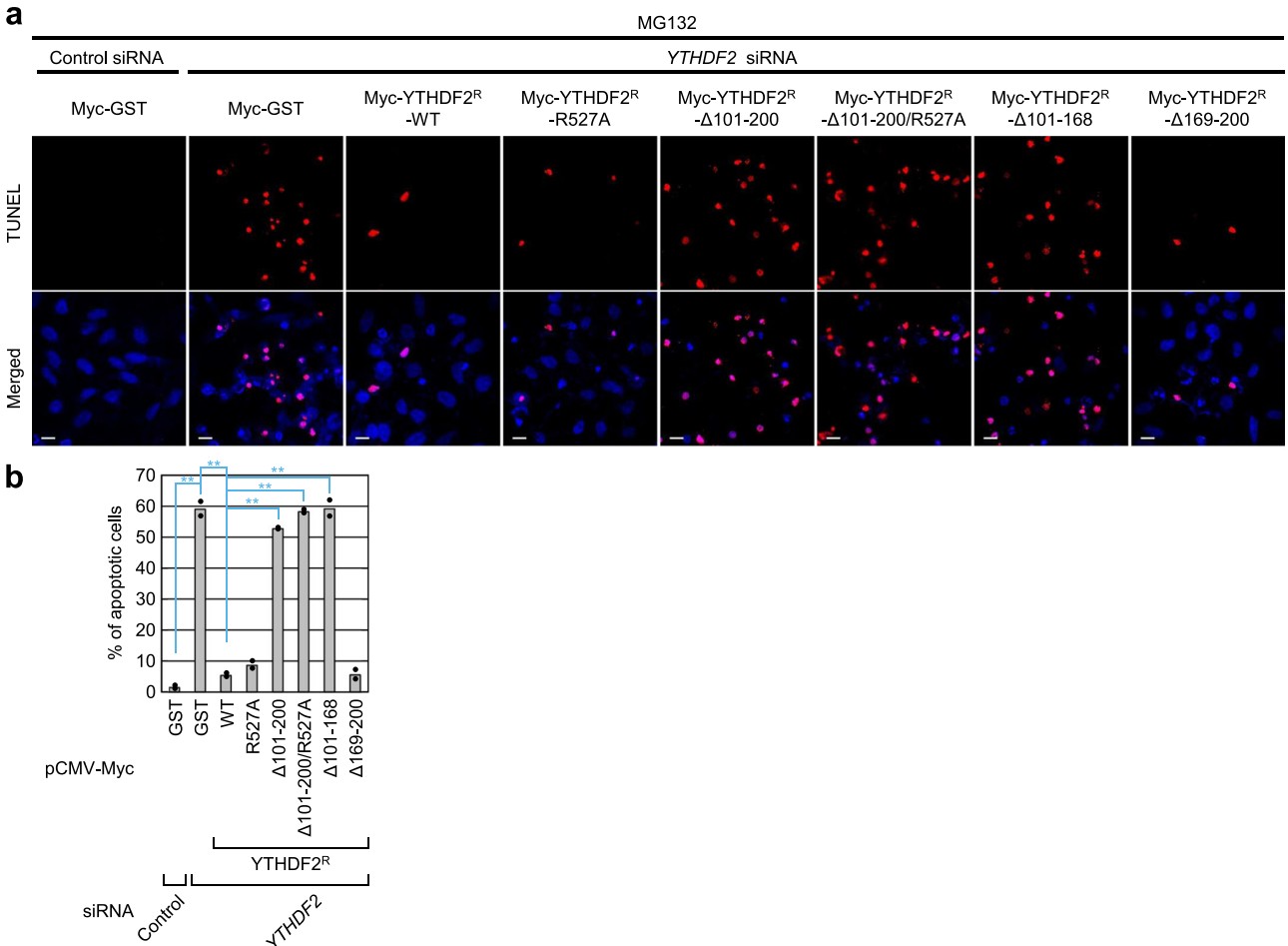

**Fig. 5 | A YTHDF2–UPF1 interaction renders cells more resistant to cellular apoptosis. a** Effect of YTHDF2 on cellular apoptosis induced by proteotoxic stresses. HeLa cells stably expressing the misfolded polypeptide CFTR-ΔF508 were either undepleted or depleted of YTHDF2. The cells were transiently transfected with a plasmid expressing Myc-YTHDF2$^R$-WT, its variant, or, as a negative control, Myc-GST. Before cell fixation, the cells were treated with either DMSO or MG132 for 16 h. Then, apoptotic cells were stained using the TUNEL staining method. Scale bar, 10 μm; $n = 2$. **b** The percentage of apoptotic cells. The stained cells in **a** were counted. The percentage of apoptotic cells is presented. ** $P < 0.01$.

## YTHDF2 is essential for aggresome formation in its condensed form

We further investigated inefficient aggresome formation upon down-regulation of endogenous YTHDF2 using another microscopy technique with a higher resolution. We expressed known misfolding-prone proteins, amyotrophic lateral sclerosis-linked G93A mutant of copper-zinc superoxide dismutase [SOD1(G93A)][31,55], synphilin1 (SYN1; which is enriched in cytosolic inclusion bodies)[31,56], and the truncated form of glutathione peroxidase 1 (GPx1-Ter)[16] in HeLa cells. All tested proteins were labeled with Dendra2, a photo-convertible fluorescent protein that allows us to observe the aggresome using stochastic localization-based super-resolution microscopy, photoactivation localization microscopy (PALM) (Supplementary Fig. 5a)[57,58]. After MG132 treatment, we observed the formation of an apparently distinct large aggresome near the nucleus for each misfolded polypeptide, along with small cytoplasmic aggregates of diffraction-limited size (<200 nm) (Supplementary Fig. 5a). In contrast, when the cells were depleted of YTHDF2, we consistently observed inefficient aggresome formation.

To compare aggresome formation patterns before and after YTHDF2 depletion, we quantitatively analyzed the localization-based super-resolved data of detected aggresomes. To this end, we defined the aggresomes using the density-based spatial clustering of applications with noise (DBSCAN) algorithm that groups signals considered to be closely packed together from the background based on two

parameters: a boundary distance and a minimum number of signals within the distance (Fig. 6a)[59]. Based on the grouped signals defined as aggresomes, we calculated the circularity of aggresomes before and after YTHDF2 depletion (Fig. 6b, c). We found that the circularity of aggresomes, measured for each misfolded polypeptide, was dramatically reduced after YTHDF2 depletion. In addition, the average number of detected signals in each aggresome was reduced in the absence of YTHDF2 for all misfolded polypeptides tested (Fig. 6d). We also observed that the number of clustered signals in the aggresome was typically one for each cell. However, under YTHDF2-depleted conditions, more than one clustered signal was detected in a single aggresome, indicating that each aggresome consisted of multiple clusters (Supplementary Fig. 5b). Collectively, our data suggest that YTHDF2 contributes to the condensation of aggresomes, forming a more circular morphology that consists of a greater number of misfolded polypeptides.

## YTHDF2 enhances the diffusion dynamics of misfolded polypeptides during movement

Given the fact that (i) YTHDF2 bridges aggregates composed of misfolded polypeptides with the CED complex and (ii) the CED complex transports the aggregates to the aggresome by interacting with dynein motor proteins, we sought to directly capture the motion of aggregates using single-particle tracking in live cells (Fig. 7 and Supplementary Movies 1–3). For this purpose, we fused HaloTag to the

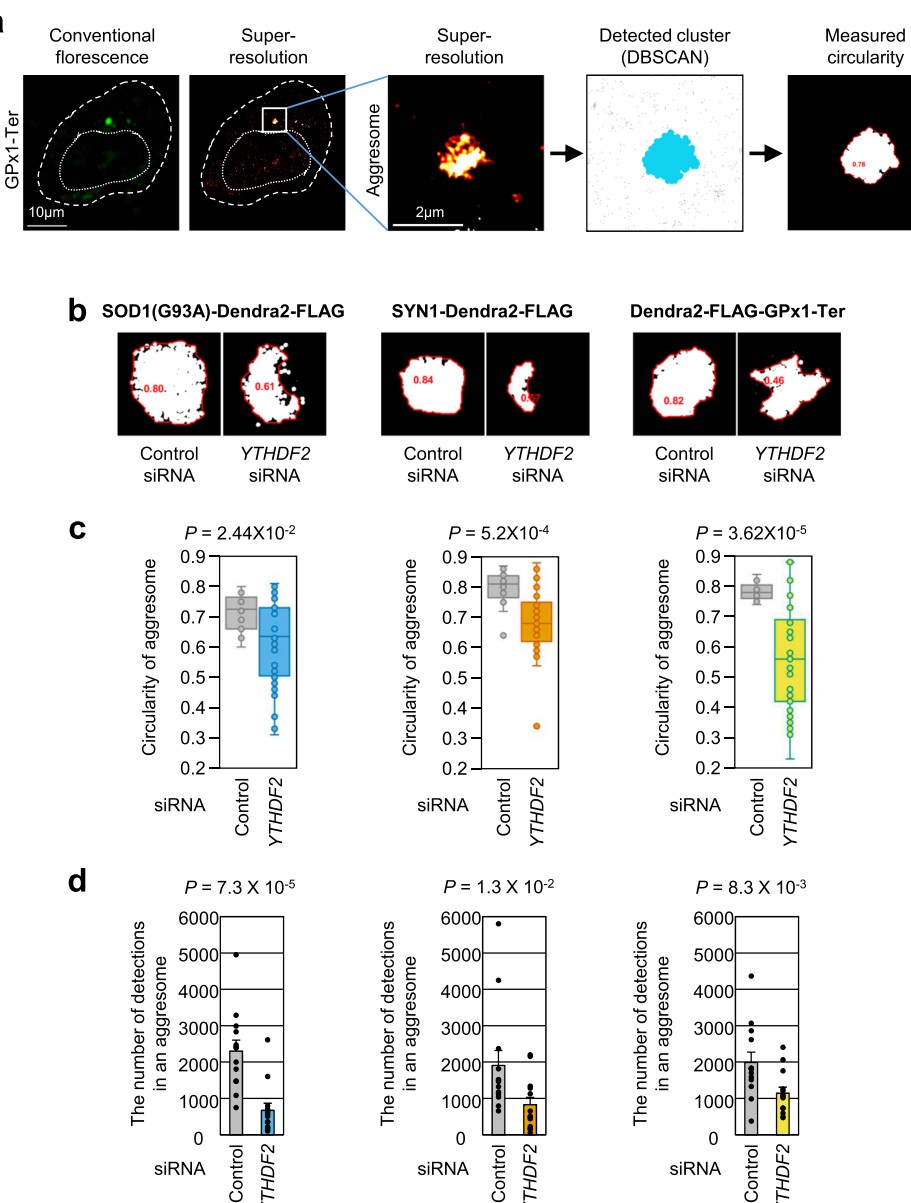

**Fig. 6 | Depletion of YTHDF2 reduces the circularity of aggresome and decreases the number of detections within the aggresome. a** Steps depicting how the super-resolution image of aggresome is reconstructed, detected as a cluster, and measured for its circularity. **b** Representative images of aggresomes with their circularities (red) detected upon control siRNA or *YTHDF2* siRNA treatment. **c** Circularity of aggresome detected in cells expressing each misfolding-prone protein. After cells were treated with MG132 for 12 h, each aggresome was imaged with super-resolution microscopy [SOD1(G93A)-Dendra2-FLAG, $n = 14$; SYN1-Dendra2-FLAG, $n = 16$; Dendra2-FLAG-GPx1-Ter, $n = 13$]. Circularity ranges from 0 to 1, and the measurement closer to 1 indicates more circular morphology. Box-whiskers show maximum, third quartile to first quartile, median and minimum. Two-tailed, equal-sample variance Student's $t$ test was used to calculate the $P$ values. **d** Relative number of detections (signals) measured within the aggresome comprising each misfolding-prone protein. Each signal represents single blinking of Dendra2 fluorescence ($n = 13$). The data from three independent experiments were analyzed and presented as mean ± s.e.m. One-tailed, equal-sample variance Student's $t$ test was used to calculate the $P$ values.

misfolding-prone proteins SOD1(G93A), SYN1, and GPx1-Ter. Since abundant expression of misfolded polypeptides can disturb the tracking of individual proteins in a cell, we sparsely labeled the HaloTag-fused misfolded polypeptides with HaloTag ligand-conjugated Janelia Flour 646 (JF646) (see Methods for more detail).

Based on our data from single-particle tracking, we calculated the mean-squared displacement (MSD) for each trajectory of the misfolded polypeptides over discrete time intervals (t), and plotted the resulting MSD values against time (MSD-t) (Fig. 7a). Each dot on the graph represents the average MSD over all trajectories within a given time interval. We observed that the increase in MSD over time intervals was significantly reduced under YTHDF2-depleted conditions

compared to under normal conditions for all misfolded polypeptides we tested.

We also analyzed the diffusion of particles by fitting the MSD-t plots with the anomalous diffusion equation for 2-dimensions, $MSD = 4Dt^\alpha$, where $D$ represents the diffusion coefficient, and $\alpha$ is the anomalous coefficient (Fig. 7a, b). We observed sub-diffusive behavior for all of the experimental conditions we tested, with no significant difference before and after YTHDF2 downregulation (Fig. 7a). However, upon the depletion of YTHDF2, the diffusion coefficients for the tested misfolded polypeptides decreased significantly (Fig. 7b). In addition, we calculated the persisted displacement for each trajectory in each experimental condition and found that the average

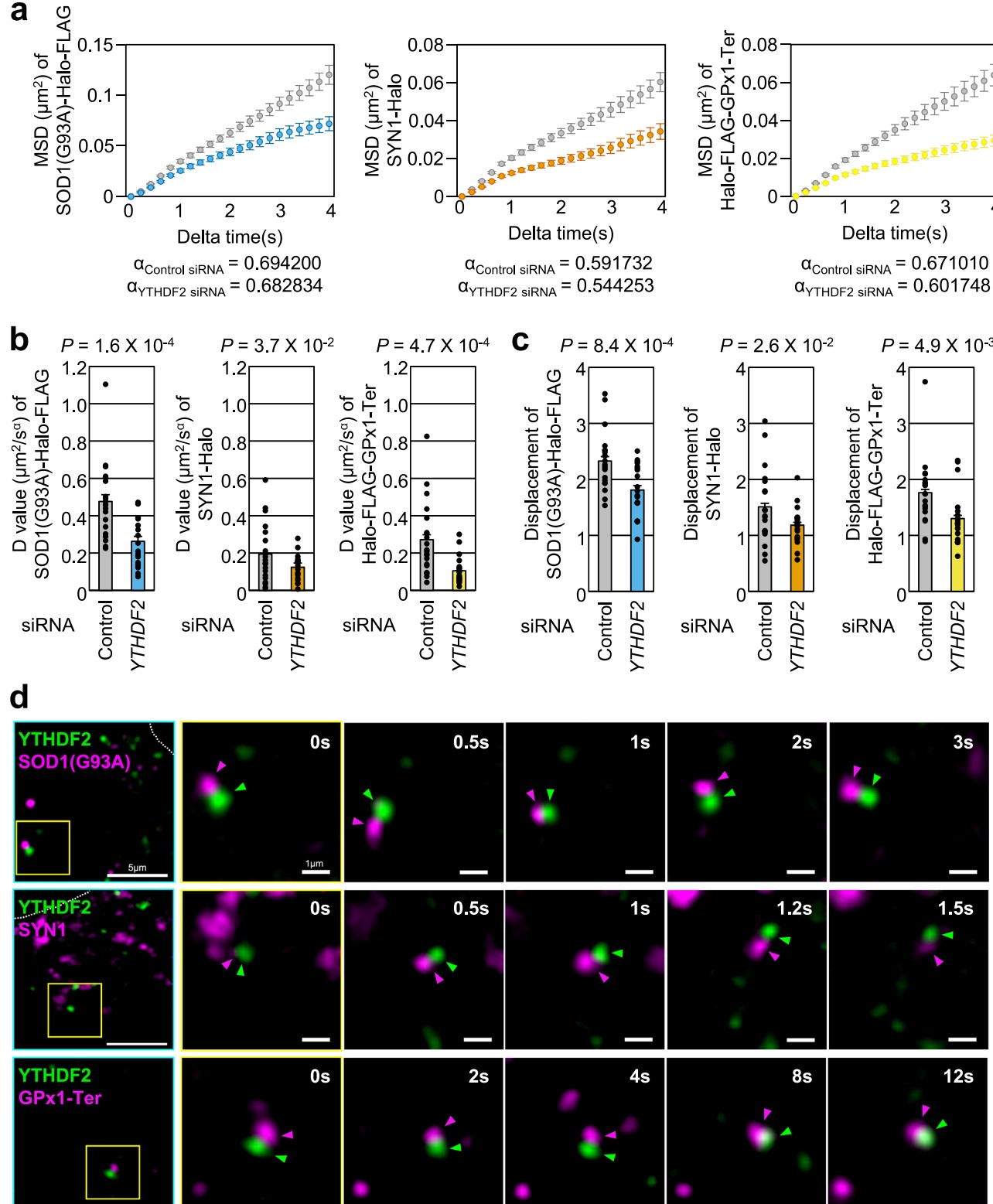

**Fig. 7 | YTHDF2 facilitates the movements of misfolding-prone proteins.**
**a** MSD-t plotted from single-particle tracking of each misfolding-prone protein. MSD values plotted in the graph were averaged data from all tracks at each time-point. Alpha values stated for each misfolding-prone polypeptide were calculated from MSD of all tracks [Control siRNA, SOD1(G93A), $n = 975$; *YTHDF2* siRNA, SOD1(G93A), $n = 765$; Control siRNA, SYN1, $n = 1077$; *YTHDF2* siRNA, SYN1, $n = 808$; Control siRNA, GPx1-Ter, $n = 1208$; *YTHDF2* siRNA, GPx1-Ter, $n = 717$]. **b** D values calculated from MSD for each misfolding-prone polypeptide. **c** Displacement of

each misfolding-prone protein. To calculate total displacement, initial position at $t = 0$ and final position of each tract were used. **d** Representative images of dual-color single-particle tracking of YTHDF2 and each misfolding-prone protein (blue box). YTHDF2 was labeled with TMR-SnapTag ligand, and misfolding-prone proteins were labeled with JF646-HaloTag ligand. Insets are time-lapse images of YTHDF2 and misfolding-prone proteins (yellow box). All data from three independent experiments were analyzed and presented as mean ± s.e.m. One-tailed, equal-sample variance Student's *t* test was used to calculate the *P* values.

displacement for each misfolded polypeptide also decreased after YTHDF2 downregulation (Fig. 7c and Supplementary Fig. 6a–c). Overall, our results suggest that YTHDF2 does not affect the anomaly of aggregate diffusion, but it does impact their speed and displacement.

Next, we attempted to verify whether misfolded polypeptides are physically associated with YTHDF2 using dual-color visualization (Fig. 7d and Supplementary Movie 4). We expressed SnapTag-fused YTHDF2 along with HaloTag-fused misfolding-prone ploypeptides in cells. To capture YTHDF2 molecules and individual misfolded polypeptides, we sparsely labeled SnapTag-YTHDF2 and HaloTag-misfolded polypeptides with tetramethylrhodamine (TMR)-conjugated SnapTag ligands and JF646-conjugated HaloTag ligands, respectively, to observe both entities in a single live cell by coincident excitation at 561 nm and 642 nm. We successfully captured the directional motion of YTHDF2, accompanied by aggregates, for all misfolded polypeptides we tested (Fig. 7d and Supplementary Movie 4). We note that the two signals were not completely co-localized, which may be due to optical aberration or the presence of a bulky protein complex between YTHDF2 and the aggregates. Nonetheless, the consistent movement of the two particles throughout the entire tracking process suggests that they are physically interacting.

We further co-expressed EGFP-fused α-tubulin with each misfolding-prone polypeptide in cells to label microtubules and observe them simultaneously using dual-color imaging. We were able to capture trajectories for all the tested misfolding-prone polypeptides, which moved towards the microtubule organizing center where aggresome is usually formed and detected (Supplementary Fig. 6d–f and Supplementary Movies 5–7). Taken together, our observations provide evidence that YTHDF2 plays an essential role in transporting aggregates and promoting the formation of the aggresome via physical interaction with misfolded polypeptides.

### YTHDF2 increases the binding of the CED-UPF1-misfolded polypeptide complex with the dynein motor protein

Considering that (i) the movement of small protein aggregates along microtubules depends on dynein motor protein[11–14] and (ii) YTHDF2 affects the speed and displacement of aggregates moving toward aggresome (Fig. 7), it is most likely that YTHDF2 may affect dynein- or dynein-mediated retrograde transport. To test this possibility, we carried out IP experiments using extracts of cells either undepleted or depleted of YTHDF2. The results revealed that YTHDF2 downregulation caused a significant decrease in the amount of co-immunopurified dynein motor protein in the IPs of polypeptidyl-puro (Fig. 8a), CTIF (Fig. 8b), or eEF1A1 (Supplementary Fig. 7). These observations indicate that YTHDF2 increases or stabilizes the association between the dynein motor protein and the complex consisting of CED, UPF1, and misfolded polypeptides, consequently facilitating the movement of dynein motor proteins for the efficient transport of small protein aggregates toward the aggresome.

## Discussion

YTHDF2 is well-characterized and intensively studied as an RNA-binding protein that specifically recognizes and modulates the stability of m[6]A-containing RNA, thereby shaping the cellular transcriptome. In this study, we provide compelling evidence that YTHDF2 contributes to protein homeostasis (Fig. 8c). We found that YTHDF2 associates with the aggresome-targeting machinery (CTIF, eEF1A1, and DCTN1) and misfolded polypeptides via its interaction with UPF1 (Fig. 3). This interaction increases the association between dynein motor protein and the complex consisting of CED, UPF1, and misfolded polypeptides (Fig. 8), promotes movement of the complex migrating along the microtubules (Fig. 7), and consequently leads to the efficient formation of the spherical shaped aggresome containing misfolded polypeptides. Notably, YTHDF2-mediated aggresome formation depends

on the UPF1-interacting ability, but not on the RNA-binding ability of YTHDF2. Collectively, our observations expand the role of YTHDF2 into PQC.

In this study, we also observed that YTHDF2 increases the circularity of aggresomes of SOD1(G93A), SYN1, and GPx1-Ter (Fig. 6), all of which form spherical aggresomes. These results suggest a possible role of YTHDF2 in the spatiotemporal distribution or condensation of misfolded polypeptides within the aggresome as well as its role in aggresomal targeting of misfolded polypeptides. In particular, given that (i) some aggresome-targeted polypeptides, such as presenilin and GFAP, form ribbon-shaped aggresomes instead of spherical aggresomes[60,61] and (ii) a mutant variant of ATP7B forms either spherical or a ribbon-shaped aggresomes depending on cell types[62], YTHDF2 may function as a discriminating factor that determines the shape of aggresomes, depending on its substrates and cell types. Furthermore, although we propose a role of YTHDF2 in aggresome formation via CED-UPF1 complex in this study, two other known aggresome-targeting cellular machineries, HDAC6 and BAG3, may interplay with YTHDF2. Indeed, a previous study suggested that CED complex associates with HDAC6 and BAG3 and acts in concert with them for efficient aggresome formation[31]. Therefore, it is most likely that YTHDF2 may contribute comparably to the circularity of aggresomes formed by an action of HDAC6 and BAG3.

The N-terminal half of YTHDF2 has a low-complexity region enriched with proline, glutamine, and asparagine. Intriguingly, this region shares a molecular feature with many proteins that are prone to undergo liquid-liquid phase separation (LLPS) to form intracellular condensates such as SGs or processing bodies[37–39]. However, it is likely that the YTHDF2-mediated formation of aggresomes containing misfolded polypeptides is mechanistically distinct from YTHDF2-driven LLPS formation in several aspects. First, all tested YTHDF paralogs (YTHDF1, 2, and 3) were comparably co-localized with G3BP1 in SGs upon treatment with sodium arsenite (Supplementary Fig. 2a). In contrast, downregulation of YTHDF2, but not of YTHDF1 and YTHDF3, specifically inhibited aggresome formation (Fig. 2). Second, while the low-complexity region spanning amino acids 230–383 in YTHDF2 is sufficient for eliciting LLPS[39], the region required for binding to UPF1 for efficient aggresome formation resides at amino acids 101–200 of YTHDF2 (Fig. 4). Lastly, while LLPS mediated by YTHDF2 is dependent on the RNA-binding ability of YTHDF2[37–39], aggresome formation was not found to depend on the RNA-binding ability of YTHDF2 (Fig. 4) or other m[6]A writers and erasers (Fig. 1). In agreement with these observations, the distribution of cellular RNAs harboring m[6]As was not significantly affected by treatment with MG132 (Supplementary Fig. 4c).

As mentioned above, the RNA-binding activity of YTHDF2 is not necessary for efficient aggresome formation. However, YTHDF2 may interact with the CED complex while bound to m[6]A RNA. Accordingly, it is expected that a subset of m[6]A RNAs may move along microtubules with the YTHDF2-CED complex and accumulate within aggresomes. Therefore, although m[6]A RNA itself does not contribute to efficient aggresome formation, m[6]A RNA accumulated in the aggresome may be subjected to either translational silencing or lysosomal degradation. In this way, YTHDF2 may coordinate the cellular transcriptome and proteome. Alternatively, a recent study revealed that the YTHDF2–UPF1 interaction drives rapid degradation of m[6]A RNAs through the 5′-to-3′ decay pathway in the cytoplasm[19]. Considering that the same interaction also contributes to efficient aggresome formation, m[6]A RNA degradation and aggresome formation may communicate with each other in a mutually exclusive or co-occurrent manner. Thus, YTHDF2 may contribute to the coordinated shaping of both the transcriptome and proteome.

Most current studies on YTHDF2 have focused on the role of YTHDF2 as a reader protein for m[6]A RNAs. However, on the basis of our present study, we would like to emphasize that many biological and

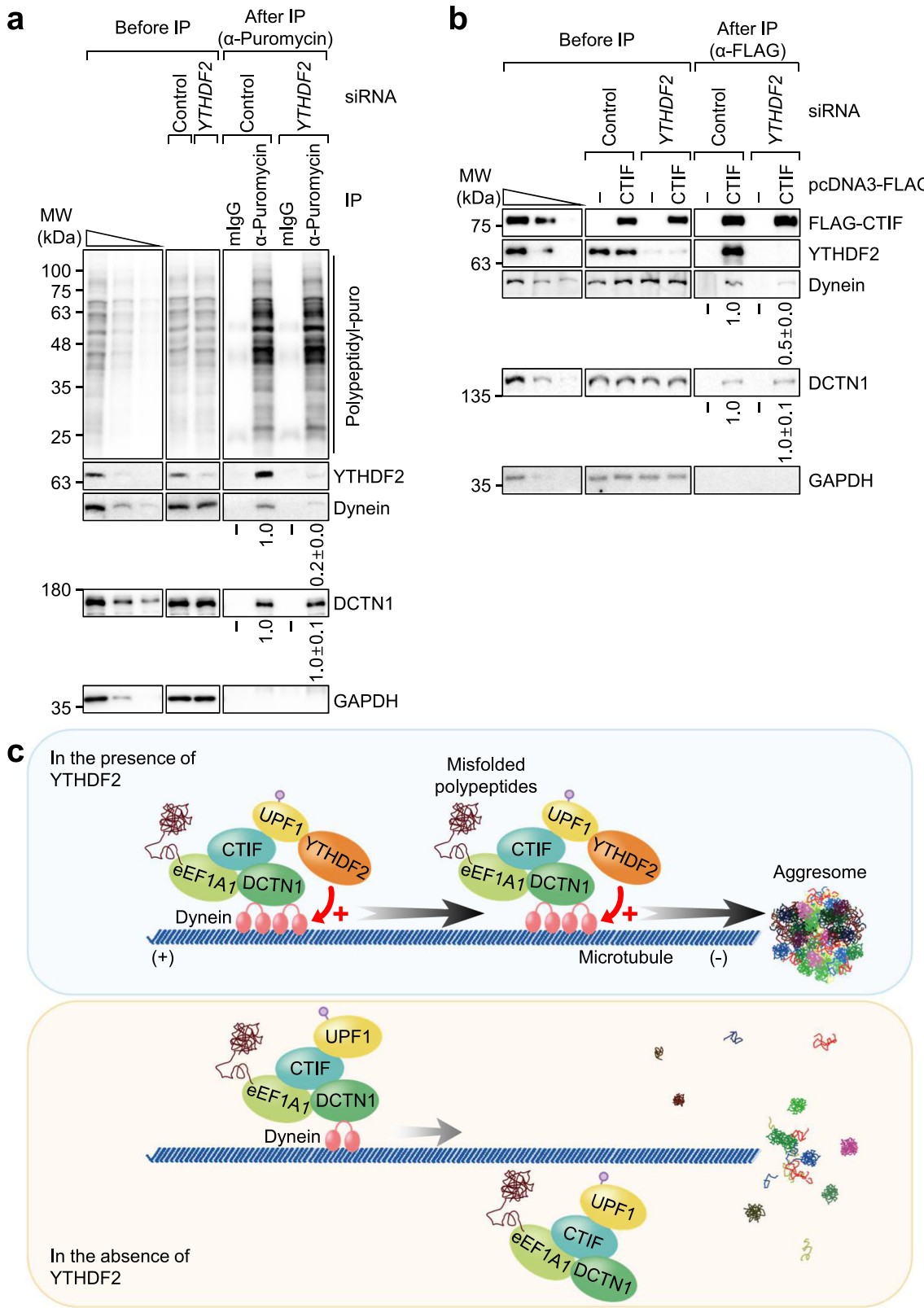

physiological events mediated by m⁶A (or YTHDF2) should be more carefully evaluated in terms of the two-fold roles of YTHDF2 as an m⁶A reader and aggresome-targeting factor. Furthermore, considering that (i) YTHDF2 plays a protective role in cell survival by promoting aggresome formation (Fig. 5), (ii) aggresomes predict poor outcomes in the pathogenesis of several cancers[63,64], and (iii) YTHDF2 is very closely associated with cancer progression[65,66], a role of YTHDF2 as an

aggresome-targeting factor as well as m⁶A reader needs more comprehensive investigation in the future.

## Methods

### Cell culture and chemical treatment

HeLa and HEK293T cells (purchased from ATCC) were cultured in Dulbecco's modified Eagle's medium (DMEM; Sigma-Aldrich)

**Fig. 8 | YTHDF2 increases an association between dynein motor protein and the CED-UPF1-misfolded polypeptide complex. a** IPs of polypeptidyl-puro. HEK293T cells were either undepleted or depleted of endogenous YTHDF2. Two days later, the cells were treated with MG132 for 12 h and puromycin for 1 h before harvesting. Then, the cell extracts were subjected to IPs using α-puromycin antibody. The cell lysate before IP was serially diluted three-fold and loaded in the three leftmost lanes to demonstrate that western blotting is semiquantitative in our experimental conditions. Relative intensities of bands in western blots of proteins obtained after IP in undepleted cells were arbitrarily set to 1.0; *n* = 3. **b** IPs of FLAG-CTIF. As performed as shown in **a**, except that the cells were not treated with puromycin and were subjected to IPs using α-FLAG antibody. Relative band intensities of proteins after IP in undepleted cells were arbitrarily set to 1.0; *n* = 3. **c** Proposed model for

YTHDF2-facilitated aggresome formation. Misfolded polypeptides tend to form small cytoplasmic aggregates, which are recognized by eEF1A1, a component of CED complex. Then, the CED complex with misfolded polypeptides recruits UPF1, which increases the frequency and fidelity of movement of the complex toward aggresome via a dynein-mediated and microtubule-dependent retrograde manner. Concomitantly, YTHDF2 is recruited to the complex via its interaction with UPF1, increases the binding of the complex to dynein motor protein and, consequently, promotes the dynein-mediated retrograde transport. The molecular mechanism by which YTHDF2 promotes the association between dynein motor protein and the CED complex with misfolded polypeptides remains unknown. See the Discussion section for more details.

supplemented with 10% fetal bovine serum (FBS; Sigma-Aldrich) and 1% penicillin/streptomycin (Sigma-Aldrich). HeLa cells stably expressing CFTR-ΔF508[31] were maintained in media containing G418 (0.4 mg/ml). Where indicated for immunostaining or IP experiments, cells were treated with either dimethyl sulfoxide (DMSO; BioShop) as a negative control or MG132 (5 μM; Calbiochem) for 12 h. Where indicated, cells were treated with puromycin (1 μg/ml for immunostaining or 10 μg/ml for IP experiments) for 1 h, sodium arsenite (500 μM; Sigma-Aldrich) for 1 h, or nocodazole (1 μM; Calbiochem) for 12 h.

For single-cell high-resolution imaging or single-particle tracking, cells were cultured on glass-bottom dishes (Cellvis, D35-25-1.5-N) at 37 °C and 5% $CO_2$ in phenol red-free DMEM (Gibco, 21063-029) supplemented with 10% FBS (Gibco, 12483-020, Canada origin, Qualified), 100 U/ml penicillin, and 100 μg/ml streptomycin (Gibco, 15140-122). Before imaging, the cell culture medium was replaced with L-15 (Leibovitz's, Gibco, 21083-027) supplemented with 10% FBS after washing with phosphate buffered saline (PBS; Gibco, 10010-023).

## Plasmid construction

The following plasmids were purchased or described previously: pCMV-Myc (Clontech #635689), pcDNA3-FLAG-YTHDF2 (Addgene #52300), pcDNA3-FLAG-YTHDF2-N (Addgene #52302), pcDNA3-FLAG-YTHDF2-C (Addgene #52303), pDendra2-N (TaKaRa #632545), and pDendra2-C (TaKaRa #632546); pcDNA3-FLAG-UPF1 and pcDNA3-FLAG-UPF1-HP[67]; p3xFLAG-GPx1-Ter, pcDNA3-FLAG-UPF1-HP-12A, and pCMV-Myc-UPF1-HP[16]; pcDNA3-FLAG, pcDNA3-FLAG-CTIF, and pmCMV-GPx1-Ter[68]; pCMV-Myc-YTHDF1, pCMV-Myc-YTHDF2, and pCMV-Myc-YTHDF3[69]; pcDNA3-FLAG-SOD1(G93A) and pCMV-Myc-GST[31]; pCMV-Myc-YTHDF2[R], pCMV-Myc-YTHDF2[R]-Δ101-168, pCMV-Myc-YTHDF2[R]-Δ101-200, and pCMV-Myc-YTHDF2[R]-Δ169-200[19]; pEGFP-N1-Synphilin1[70]; and pHAGE-Ubc-NLS-MCP-HaloTag[71].

To construct pcDNA3-FLAG-YTHDF2[R]-R527A, an NheI/XhoI fragment from pcDNA3-FLAG-YTHDF2 was ligated to the NheI/XhoI fragment from a two-step PCR product. The 1st PCR was performed using pcDNA3-FLAG-YTHDF2 as a template and two pairs of oligonucleotides: (i) 5′-GTTTGCCTCCAGCCACCATTGCTCCTCC-3′ (sense) and 5′-GTGTCCGCAGAGTTGGTCACTGGTTTATTC-3′ (antisense); (ii) 5′-GTGACCAACTCTGCGGACACTCAGGAAGTG-3′ (sense) and 5′-GATCAGCGAGCTCTAGCATTTAGGTGACAC-3′ (antisense). The 2nd PCR was performed using the product of the 1st PCR as a template and two oligonucleotides: 5′- GTTTGCCTCCAGCCACCATTGCTCCTCC-3′ (sense) and 5′- GATCAGCGAGCTCTAGCATTTAGGTGACAC-3′ (antisense).

The plasmid pCMV-Myc-YTHDF2[R]-R527A was generated by ligating a HindIII/XhoI fragment of pcDNA3-FLAG-YTHDF2[R]-R527A to a HindIII/XhoI fragment of pCMV-Myc-YTHDF2.

To construct pCMV-Myc-YTHDF2[R]-Δ101-200/R527A, a HindIII/Klenow-filled EcoRI fragment of pCMV-Myc-YTHDF2[R]-R527A was ligated to a HindIII/Klenow-filled Acc65I fragment of pcDNA3-FLAG-YTHDF2(Δ101-200).

To construct pSynphilin1-Halo, the GFP fragment in pEGFP-N1-Synphilin1 was replaced with a PCR-amplified fragment containing a Halo tag from pHAGE-Ubc-NLS-MCP-HaloTag[71].

To construct pSOD1(G93A)-Halo-FLAG, a fragment encoding SYN1 and Halo in pSynphilin1-Halo was replaced with two fragments. One fragment containing SOD1(G93A) was obtained through PCR using pcDNA3-FLAG-SOD1(G93A)[31] and the other fragment containing Halo-FLAG was amplified through PCR using pSynphilin1-Halo.

To construct pHalo-FLAG-GPx1-Ter, GFP and HP1α fragments in GFP-HP1α (Addgene #17652) were replaced with two fragments: a PCR-amplified fragment containing a Halo from pHAGE-Ubc-NLS-MCP-HaloTag and a PCR-amplified fragment containing FLAG-GPx1-Ter from pmCMV-GPx1-Ter.

To construct pSOD1(G93A)-Dendra2-FLAG and pSynphilin1-Dendra2-FLAG, a fragment containing SOD1(G93A) amplified through PCR using pcDNA3-FLAG-SOD1(G93A) as a template and a fragment containing SYN1 amplified through PCR using pSynphilin1-Halo as a template, respectively, were inserted upstream of pDendra2-N.

To construct pDendra2-FLAG-GPx1-Ter, a fragment encoding GPx1-Ter from p3xFLAG-GPx1-Ter was inserted into pDendra2-C.

To construct pSnapTag-Myc-YTHDF2[R], a fragment containing SnapTag amplified through PCR using MCP-NLS-SnapTag[72] as a template was inserted into pCMV-Myc-YTHDF2[R].

## DNA or siRNA transfection

For DNA transfection, HEK293T cells were transiently transfected using the calcium-phosphate method for IP experiments, and HeLa cells were transfected using Lipofectamine 2000 (Invitrogen) or jetPRIME (Polyplus) for confocal microscopy experiments.

For siRNA transfection, the cells were transfected with 100 nM in vitro-synthesized siRNA using Lipofectamine 3000 (Invitrogen). The following siRNA sequences were used in this study: 5′-r(CCGCGUCUAGUUGUUCAUGA)d(TT)−3′ for human *YTHDF1*, 5′- r(AAGGACGUUCCCAAUAGCCAA)d(TT)−3′ for human *YTHDF2*, 5′- r(AUGGAUUAAAUCAGUAUCUAA)d(TT)−3′ for human *YTHDF3*, 5′- r(CAGAGAAGGCAGAACUAAA)d(TT)−3′ for human *DCTN1*, 5′- r(AAGCCAGUGAUACAGCGAAUU)d(TT)−3′ for human *SMG6*, 5′- r(GCAAGUAUGUUCACUAUGA)d(TT)−3′ for human *METTL3*, 5′- r(GGAUGAGUUAAUAGCUAAA)d(TT)−3′ for human *METTL14*, 5′-r(AUAGCCGCUGCUUGUGAGA)d(TT)−3′ for human *FTO*, and 5′- r(GAUGCAGUUCCGCUCCAUU)d(TT)−3′ for human *UPF1*. The sequences of nonspecific control siRNA were previously described[73].

## Antibodies

Antibodies against the following proteins (for immunostaining, western blotting, or IPs) were used in this study as described previously: SMG6[74], UPF1[75], and CTIF[68].

The purchased antibodies against the following proteins are listed in the format "protein name (catalog number, supplier)": FLAG (DYKDDDDK; 14793, Cell Signaling Technology or A8592, Sigma-Aldrich), Myc (9E10; OP10L, Calbiochem or 2272, Cell Signaling Technology), FTO (ab124892, Abcam), GFP (sc-9996, Santa Cruz Biotechnology), DCTN1 (p150glued; 610474, BD Biosciences), eEF1A1 (CBP-KK1; EF1α; 05-235, Merck Millipore), YTHDF1 (17479-1-AP, Proteintech), YTHDF2 (24744-1-AP, Proteintech), YTHDF3 (sc-377119,

Santa Cruz Biotechnology), METTL3 (15073-1-AP, Proteintech), METTL14 (HPA038002, Sigma-Aldrich), p-(S/T)Q ATM/ATR substrate (2851, Cell Signaling Technology), m6A (#202003, Synaptic Systems), m1A (D345-3, MBL), puromycin (12D10; MABE343, Merck Millipore), β-actin (A5441, Sigma-Aldrich), GAPDH (LF-PA0212, AbFrontier), α-tubulin (sc-53030, Santa Cruz Biotechnology), γ-tubulin (sc-17788, Santa Cruz Biotechnology), dynein (sc-9115, Santa Cruz Biotechnology), G3BP1 (13057-2-AP, Proteintech), IMPβ (A301-803A-1, Bethyl Laboratories), Alexa Fluor 488 goat α-mouse IgG (A-11017, Invitrogen), and rhodamine-conjugated goat α-rabbit IgG (31670, Invitrogen).

### Western blotting
Harvested cells or IP samples were incubated with 2× sample buffer [100 mM Tris-HCl (pH 6.8), 4% SDS, 10% β-mercaptoethanol, 15% glycerol, and 0.008% bromophenol blue] for 5 min at 95 °C. Protein samples were separated by sodium dodecyl sulfate-polyacrylamide gel electrophoresis and transferred to a Protran Premium nitrocellulose membrane (Amersham). The membranes were incubated with primary and secondary antibodies to detect specific proteins. Protein signals were quantitated using Amersham Imager 600 (GE Healthcare) or Amersham ImageQuant 800 (GE Healthcare).

### Immunoprecipitation
HEK293T cells were harvested and resuspended in NET-2 buffer [50 mM Tris-HCl (pH 7.4), 150 mM NaCl, 1 mM phenylmethylsulfonyl fluoride (Sigma-Aldrich), 2 mM benzamidine hydrochloride (Sigma-Aldrich), 0.05% NP-40 (IGEPAL CA-630; Sigma-Aldrich), 10 mM sodium fluoride (Sigma-Aldrich), and 0.25 mM sodium orthovanadate (Sigma-Aldrich)]. The suspended cells were sonicated and pre-cleared with protein A or G agarose 4 B beads (Incospharm) for 1 h at 4 °C. The pre-cleared samples were incubated with FLAG M2 affinity gel (Sigma-Aldrich) or antibody-conjugated beads for 3 h at 4 °C. The beads were washed four times with NET-2 buffer, and the bead-bound proteins were eluted using 2× sample buffer. The samples were analyzed by western blotting.

Where indicated, cell extracts before IP were treated with RNase A. Total RNAs were purified with TRIzol Reagent (Invitrogen). Then, purified total RNA samples were mixed with in vitro-transcribed FLuc RNAs as a spike-in. The amounts of mRNAs were quantified using quantitative reverse-transcription PCR (qRT-PCR), and the level of endogenous *GAPDH* mRNAs were normalized to FLuc RNAs. 5′-TG GCAAATTCCATGGCACC-3′ (sense) and 5′-AGAGATGATGACCCTTT TG-3′ (antisense) oligonucleotides for amplification of *GAPDH* mRNAs and 5′-CAACACCCCAACATCTTCG-3′ (sense) and 5′-CTTTCCGCCCT TCTTGGCC-3′ (antisense) oligonucleotides for amplification of FLuc RNAs.

### Immunostaining
HeLa cells were fixed with 3.65–3.8% formaldehyde (Sigma-Aldrich) and permeabilized with 0.5% Triton X-100 (Sigma-Aldrich). The cells were then incubated with 1.5% bovine serum albumin (BSA; BovoStar) for 1 h and incubated with primary antibodies in 0.5% BSA for 1 h. Next, the cells were incubated with secondary antibodies conjugated to Alexa Fluor 488 or rhodamine in 0.5% BSA for 1 h. Nuclei were stained with 4′,6-diamidino-2-phenylindole (DAPI; Biotium) for 5 min. Immunostained cells were observed using LSM 510 Meta, LSM 700 confocal microscope (Carl Zeiss), and ECLIPSE Ti2-E (Nikon).

### Microtubules/tubulin in vivo assay
The relative amounts of microtubule and free α/β-tubulin in cells were determined using the Microtubules/Tubulin In Vivo Assay Kit (cytoskeleton, Cat. # BK038), following the manufacturer's instructions. In brief, HeLa cells depleted of either UPF1 or YTHDF2 were lysed in a microtubule stabilization buffer. After cell lysis, total-cell extracts were

centrifuged at $1000 \times g$ for 5 min. The supernatant was then centrifuged at $100,000 \times g$ for 1 h to separate polymerized microtubules (pellet fraction) and free α/β tubulin (supernatant fraction). The fractionated samples were then analyzed by western blotting using an α-tubulin antibody.

### Highly inclined and laminated optical (HILO) illumination
To reduce the background noise and increase the signal-to-noise ratio for fluorescence detection, we used HILO illumination, which was transmitted in the form of a thin sheet with an incident angle smaller than the critical angle for total reflection. Cells for high-resolution imaging or single-particle tracking were exposed to HILO illumination from a custom-built microscope based on a Nikon microscope body (Ti2e) with 405-nm, 488-nm, 531-nm, and 640-nm excitation lasers (CUBE, Coherent). Sequential images from live cells were acquired through a 100X objective lens of 1.49 numerical aperture (Nikon, Plan Apochromat TIRF 100X oil-type) with an electron-multiplying charge-coupled device (Andor, iXon Ultra 897), and processed using the NIS-Elements software (Nikon). Image analysis was performed using ImageJ scripts and custom-built MATLAB codes.

### Photoactivated localization microscopy (PALM) imaging
For PALM imaging of Dendra2, cells were illuminated with 405-nm (for photoconversion of Dendra2) and 561-nm laser (for excitation of photoconverted Dendra2)[57]. Each image was acquired at a temporal resolution of 50 ms/frame until most of the photoconverted Dendra2 signals were bleached in the nucleus. The acquired images were analyzed and reconstructed into super-resolved images using MTT[76] and qSR[77].

### Density-based spatial clustering of applications with noise (DBSCAN) analysis
We defined clusters in the aggresome on super-resolved images using the DBSCAN algorithm[59]. We investigated groups of each detected molecule based on single-molecule localization. Each group was defined as a single cluster using DBSCAN. For DBSCAN, we used two parameters: the minimum number of detections in a group ($N$), and the minimum distance between detections in a group ($R$). To define aggresome clusters, we used $N = 10$ points and $R = 200$ nm [for SOD1(G93A)-Dendra2-FLAG and -Dendra2-FLAG-GPx1-Ter] or $R = 230$ nm (for SYN1-Dendra2-FLAG).

### Circularity analysis
From the clusters defined within the aggresome using DBSCAN, we investigated the circularity of each cluster using a custom MATLAB script. Based on each of the pointillist results from super-resolution imaging, the boundary of each aggresome was plotted with a line. The detected boundary was then estimated as a circular object to compute its perimeter and area. The circularity value was calculated using the roundness metric formula.

### Single-particle tracking
The ImageJ plugin software TrackMate was used for single-particle tracking, and custom-built MATLAB codes were used for the MSD calculation. Fluorescent particles per frame were identified using Laplacian Gaussian fitting with a log detector in TrackMate. For each particle, two-dimensional trajectories of the X-Y coordinates over time were obtained.

### MSD calculation
For the aggregate or YTHDF2 movement analysis, single-particle analysis was performed using custom-built MATLAB codes. MSD was calculated using a method previously reported[78], where the MSD($n\Delta t$) for a given time lag ($n\Delta t$) is defined as the average over all points for the

given time lag:

$$\text{MSD}(n\Delta t) = \frac{1}{N-n} \sum_{i=1}^{N-n} [x(i+n) - x(i)]^2 + [y(i+n) - y(i)]^2$$

where $N$ represents the length of the trajectory, and $\Delta t$ is the time delay between frames. To estimate the diffusion coefficient ($D$) and anomalous diffusion parameter ($\alpha$), MSD-t plots were fitted using $4Dt^{\alpha}$ for two-dimensional motion.

**Fluorescent tag ligand incubation for single-particle tracking**
For HaloTag- or SnapTag-labeled single-particle imaging, JF646-HaloTag ligand (Janelia Fluor HaloTag Ligands, Promega, GA1120) and TMR-SnapTag ligand (SNAP-Cell TMR-Star, NEB, S9105S) diluted in DMSO were added to the cell culture media and incubated at 37 °C and 5% $CO_2$. Non-bound ligands were removed after a 15 min incubation with cell culture media, which was repeated twice. For single-particle tracking, we sparsely labeled the protein tags with 1 nM for 7 min for each ligand, which is 100 times more diluted than the usual protocol.

For dual-color single-particle tracking of misfolded polypeptides and YTHDF2, OptoSplitII (Cairn Research) was used to simultaneously image two different wavelengths. To improve the detected signals, we labeled the protein tags with slightly higher concentrations: 20 nM for TMR-SnapTag and 10 nM for JF646-HaloTag.

**Statistical analysis**
Most of the data were obtained from three independently performed biological replicates. Two-tailed and equal-variance Student's $t$ tests were performed to compare the data. All data are presented as mean ± standard deviation, and statistical significance was defined as a $P$ value $< 0.05$ or $<0.01$.

For IP experiments, the average values of three biological replicates were calculated and presented with standard deviations. Box plots were used to represent the data, and one-way ANOVA test was used to calculate the $P$ values. The box-whisker plot displays the maximum, third quartile to first quartile, median, and minimum values.

To quantify the populations of cells exhibiting aggresomes, more than 150 cells obtained from three independent biological replicates were examined. Two-tailed equal-variance Student's $t$ test was used to calculate the $P$ values, and at least three experienced independent investigators counted cells in a blinded manner.

**Reporting summary**
Further information on research design is available in the Nature Portfolio Reporting Summary linked to this article.

## Data availability
The data supporting the findings of this study are available from the corresponding authors upon request. Source data for the figures and supplementary figures are provided as a Source Data file. Source data are provided with this paper.

## Code availability
All codes used for circularity and statistical analyses are available on GitHub (https://github.com/wonkicholab/YTHDF2-Analysis).

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

## Acknowledgements

This work was supported by a National Research Foundation (NRF) of Korea grant funded by the Korean government (Ministry of Science, ICT, and Future Planning; NRF-2015R1A3A2033665, NRF-2018R1A5A1024261, and NRF-2022M3E5F1017965 to Y.K.K.; RS-2023-00208539 and NRF-2022M3A9B6017424 to W.-K.C.; and RS-2023-00242239 to H.J.H.) and the Suh Kyungbae Foundation (to W.-K.C.).

## Author contributions

H.J.H., T.L.P., H.-I.K., W.-K.C. and Y.K.K. conceived and designed the experiments. H.J.H., H.-I.K and Y.P. performed the biochemical experiments. T.L.P., G.K. and C.S. performed the single-particle experiments and analyzed the imaging data under the supervision of W.-K.C. H.J.H., T.L.P., H.-I.K., W.-K.C. and Y.K.K. wrote the manuscript.

## Competing interests

The authors declare no competing interests.
