## [Peer Review File · Nature Communications]

YTHDF2 facilitates aggresome formation via UPF1 in an m6A-independent mannerReviewers' Comments:

Reviewer #1:

Remarks to the Author:

Summary

This is a nice study where the authors report that in HeLa cells, the N6-methyladenosine (m6A) RNA-binding protein YTHDF2 plays a role in protein quality control. Specifically, they find that YTHDF2 plays a role in aggresome formation through its interaction with UPF1 and the misfolded proteins targeted for degradation. UPF1 is required for YTHDF2's interaction with the CED complex (CTIF, eEF1a1, and DCTN1). YTHDF2 is required to increase the interaction between dynein and the misfolded polypeptide-associated CED complex and UPF1 needed for directed movement to the centrosome and formation of the aggresome. Downregulation of YTHDF2 leads to no aggresome and increased apoptosis. YTHDF2's RNA binding activity was not required for these effects, and this function was not due to the intrinsically disordered domain, but specifically to its interaction with UPF1.

General

The authors addressed the question of how YTHDF2 plays a role in protein turnover in HeLa cells through well-designed experiments, and many different types of methodologies. The data are clear, and visually pleasing, and largely support the claims of the authors. The authors further support their findings with multiple different approaches including both endogenously labeled misfolded proteins, as well as overexpressed proteins known to aggregate. It was also well-written, and easy to read and follow. There are quite a lot of figures, but they are large and clear, and everything is legible, which is much appreciated. They used very detailed approaches to answering their questions with overexpression, knockdown, mutations, and depletions. My biggest weakness is cell type and relevance. Because there have been recent papers showing that use of the aggresome is cell-type specific, how well these findings would apply to work in more relevant cell types is unclear. How much other pathways to aggresome formation (HDAC6, BAG3, etc.) apply in these cells is also unclear, and may want to be added in the discussion. The ease of use in working with the heavily mutated HeLa cells does allow for many experiments to be done that might not be possible in primary cell types. Perhaps the authors may want to make some type of discussion about aggresome formation and cancer, since their work suggests that in a cancer line (HeLa cells), lack of aggresome formation leads to apoptosis, showing that it would be protective. If there were any other relationship they could show to support their findings in other cancer cells or lines would also be helpful and provide relevance. I have many comments about the manuscript below, however, I recommend to accept with minor revisions required.

Major

- None

Minor

- I have difficulty accepting the word "dispersed." Typically dispersed suggests that things were once together but now are no longer so, such as "the crowd dispersed." Because most of the data are fixed endpoints, it is not clear if the "dispersed" phenotype occurred because things didn't get to the aggresome, or that they were at the aggresome and then it fell apart (which is what most will intuit with the word "dispersed"). Unless specific data are done such as a timelapse supporting the latter situation, the word should be changed.

- Please add that the cells you are working with are HeLa cells in the abstract and early in the Results section as it was not clear until much later.

- In the introduction, the authors say that "aggresomes" are similar to inclusion bodies, which is contrary to specific studies showing that specific cargoes go to inclusion bodies versus aggresomes; aggresomes are soluble compartments (JUNQ), whereas inclusion bodies are insoluble (iPOD) (Kaganovich et al, 2008 Nature; Ogrodnik et al, 2014 PNAS) Please modify.

- Reference 15, which is used throughout the introduction for citation of important statements about aggresomes is a preprint written by an author at a community college who has not studied the aggresome (per Pubmed). This is not acceptable for supporting such critical statements. Use a different citation.
- Knockdown of METTL3 and METTL14 was not very complete per western blot. Can you confirm that no RNAs in those conditions are m6A-modified? Statement in the text says "...in contrast, downregulation of the m6A writer...marginally affected aggresome formation." There is barely a reduction of METTL3 and 14 following knockdown and it was not quantified. The authors cannot make that claim unless they accurately describe in the text that that percentage knockdown did not affect aggresome formation. Alternatively, better knockdown could be attempted. As is, there's not much that can be claimed when there is so little knockdown. Further, METTL14 appears to also be reduced with METTL3 knockdown. Again, quantification of the knockdown should be reported.
- For all of the graphs such as in Figure 1b, Fig2 b,d, etc., authors should report what the DMSO treated condition looked like with all of these knockdowns to suggest if the phenotype is specific to show.
- Fig 1 legend says all data represent at least 2 biological replicates, n=2. These are HeLa cells. If they were primary cells that were hard to obtain and work with, maybe that might be OK, but there should be minimally an n of 3 for this work.
- Fig S4B should be put into the main figure. This finding is important.
- The movies, while very cool, don't suggest microtubule movements. The movies are short, with no directionality, is there movement towards the centrosome? This looks like a fence and picket type movement as is.
- Statement in line 356 suggests that there is direct evidence that YTHDF2 plays a role in recruiting aggregates to the CED-motor protein complex, leading to the formation of the aggresome. What you have shown in this paper is interaction but no directionality of recruitment or movement towards the aggresome.
- Please put the information in lines 866-868 about total lysate and what those gradients stand for in the western blots in every figure legend with that serial dilution (for e.g. Fig 1)
- Put Fig 4C into the supplement
- In the image in Fig 8C, the location of YTHDF2 doesn't really represent what the authors suggested for its requirement in the co-IP where dynein was detected (Fig 8). How better might this be represented to show that relationship?

Reviewer #2:

Remarks to the Author:

The manuscript of Hwang et al investigates the role of YTHDF2 in aggresome formation. Overall, the work is interesting but some explanations and clarifications as needed before the manuscript can be considered for publication.

Major points:

1) Figures 1, 2 and 4. It is not clear to me what the authors are evaluating here, which makes it difficult to judge the rest of the work. They are comparing distinct aggresomes versus dispersed or

lack of aggresomes per cell. What is the definition of a "dispersed aggresome"? From a visual inspection of the images, I do not reconstruct the bar graphs (1b, 2b, 2d and 4b) in these figures. The authors need to clearly explain to the readers how they exactly perform this evaluation (is it the brightness, size, number, of the puncta that indicate aggresomes? or what information is used) so that the reader can follow the paper.

2) Page 13: "The super-resolved structure of the aggresome was not spherically shaped, implying that the aggresome is not in liquid phase". The authors should provide appropriate controls to demonstrate that the non-spherical shapes are not an artifact of the fixation procedure.

3) Page 15 and Figure 7. There are many inconsistencies with the single particle tracking analysis that should be clarified.

a) Line 333. "We noted that the α value for SYN1 was below zero," First of all, I do not see an α value of less than zero in Figure 7a. Do the authors mean "below 1"? If α is less than zero, then can be do to either an incorrect fit, or too many data points used in the MSD calculation (as a rule of thumb, one does not calculate shifts in the MSD larger than 1/4 of the trajectory length, and sometimes one needs to use even shorter shifts for the MSD).

b) Line 335. "resulting in freely diffusive SYN1 aggregates in our measured population." If we are referring to the MSD of the SYN1 in Figure 7a, then one has subdiffusion (i.e. $\alpha < 1$ or anomalous diffusion) and not free diffusion.

c) Figure 7a. It would be useful to show the MSD fits in the plots to get an idea of how descriptive the parameters are. Also, an α value of 1.6 in the right most panel does not make sense when considering the entire plot. The MSD looks more linear, implying an $\alpha = 1$. Only the early times (first five data points) show directed motion. Also, in panel 7b, the units are incorrect. The diffusion coefficients should be given in " $\mu\text{m}^2/\text{s}^\alpha$ ".

d) Lines 337-339. "It should be noted that, upon depletion of YTHDF2, both the anomalous coefficients and diffusion constants for the tested misfolded polypeptides reduced drastically (Fig. 7a,b)". Actually, this statement is not exactly precise. Indeed, the exponent decreases indicating stronger subdiffusional or anomalous behavior. However, the diffusion coefficient for SOD and SYN1 are pretty much unchanged (when looking at the diffusive part of the MSD plot within the first ~ 1 s). It is true that there is a drastic reduction in the diffusion coefficient of the Halo-FLAG-GPx1-Ter.

e) Figure 7d. The co-diffusion is confusing. In all cases, the spots are separated by $\sim 1 \mu\text{m}$. If the particles are moving together, I would expect them to be closer together. The authors should explain what they are observing here in more detail.

4) Lines 353-355. "We successfully captured the directional motion of YTHDF2, accompanied by aggregates, for all the misfolded peptides we tested (Fig. 7d and Supplementary Video S1d)." The authors need to substantiate this statement better. There is no evidence in Figure 7d and Supplementary Video S1d that the motion is directional nor is the evidence for co-diffusion very strong.

Minor points:

1. Line 141-143: "Immunostaining revealed that either FLAG-YTHDF2 or endogenous YTHDF2 was evenly distributed throughout the cytoplasm regardless of MG132 treatment (Fig. 1c and Extended Data Fig. 1d)." I have difficulties understanding the "either" in this sentence. That would imply that sometimes FLAG-YTHDF2 is evenly distributed, and sometimes YTHDF2, but never both at the same time, which does not make sense to me. Please reword the sentence for readability.

2. Overall, the English is fair in the paper. However, there are a few places where the English detracts from the readability of the text.

Reviewer #3:

Remarks to the Author:

This manuscript from the Kim laboratory investigates potential connections between the m6A reader protein YTHDF2, the RNA decay factor UPF1, and protein quality control. The authors propose that YTHDF2, uniquely among YTHDF paralogs, is important for formation and morphology of aggresomes

formed under a high burden of misfolded proteins. The authors provide evidence that depletion of YTHDF2 impairs the coalescence of dispersed protein aggregates into aggresomes, which they link to increased sensitivity to proteotoxic stress. Loss of YTHDF2 is also associated with an apparent reduction in the interaction between the CED complex and dynein, which in turn impairs transport of aggregates to the aggresome. Based on co-IP and knockdown experiments, the authors develop a model where UPF1 recruits YTHDF2 to the CED complex. The mechanism by which YTHDF2 association with the CED complex promotes aggresome formation is undefined. The possibility of links between RNA and protein quality control pathways is intriguing, but I have many reservations about the manuscript as currently presented. The significance and regulatory rationale of the involvement of YTHDF2 in aggresome formation is unclear, and little mechanistic insight is provided. I also have concerns about data quality and interpretation. Please see below for detailed comments.

Major points:

1. Figure 1 and S1C: The authors use METTL3 and METTL14 knockdown to ask whether the effect of YTHDF2 on aggresome formation is dependent on m6A. However, S1C shows that substantial amounts of both m6A writer proteins persist in the knockdown conditions. Given the presence of METTL3/14 in the assayed cells, it is inappropriate to use these data to conclude that the role of YTHDF2 is m6A-independent.
2. In Figure 1, 2, and 4, the authors use IF to examine aggresome formation, and these are key pieces of data for their argument. However, it is not clear to me how aggresome formation was quantified in these experiments. The Y axes of the graphs in 1B, 2B, 2D, 4B say "Relative distribution of [polypeptidyl puoro or CFTR-deltaF508]" but the figure legends say that this reports the "Relative percentages of cells containing either aggresomes or dispersed aggregates." I don't think these descriptions are interchangeable (the axis labels imply quantification of signal inside and outside of aggresomes, which would be preferable). I also don't understand what is meant by "relative percentage" in these figure legends. Most importantly, I could not find a description of the criteria used to distinguish an aggresome from dispersed aggregates. This is critical for interpretation of the data.
3. Figure 2A, the overall signal in YTHDF3 cells appears lower. What accounts for this and how might it affect the quantification?
4. Figure S2B: It is not possible to exclude that YTHDF2 knockdown interferes with microtubule organization based on these images in the absence of quantification.
5. Throughout the quantification of the westerns is questionable. Virtually every standard deviation is +/- 0.0 or 0.1, which is unlikely if the experimental error is normally distributed.
6. Figure 4A: Does the expression of wild-type YTHDF2 faithfully rescue aggresome morphology in this assay? These aggresomes don't look like those in the control and the overall signal is faint. If this is a representative image, I have doubts about the validity of the knockdown-rescue system used. As above, the question of how aggresomes vs aggregates were quantified is essential.
7. Figure 7: I don't understand the basis for this statement, given the data shown: "Collectively, our observations from the single-particle tracking analysis indicate that YTHDF2 enhances the processivity of misfolded polypeptide movements and the average moving distance of small aggregates per association with the microtubule without detaching." Did the authors perform analyses of misfolded polypeptide processivity in the presence and absence of YTHDF2? I don't think that the aggregate measures of displacement shown in figure 7 give direct insight into processivity.
8. Figure 7: It's not clear to me why YTHDF2 would have an effect on SYN1 displacement that is similar to the other two model misfolded proteins, since the authors seem to be suggesting that the

amount of free SYN1 protein limits the ability to assay directed motion/assisted diffusion: "We noted that the α value for SYN1 was below zero, which may be due to excessive overexpression of the protein compared to capable motor proteins or CED complex, resulting in freely diffusive SYN1 aggregates in our measured population." Can the authors explain further?

Minor points:

1. How do the differences in aggresome shape fit into the model? A discussion of this would be helpful, since that phenotype was highlighted in the abstract.
2. Figure 1C: The significance of finding FLAG-YTHDF2-N in aggresomes following MG132 treatment isn't clear to me. "These data suggest that the N-terminal half of YTHDF2 has molecular features associated with protein misfolding or aggresome formation." Is a very vague statement. This could be said of any misfolded protein and does not give insight into the function of YTHDF2.
3. Also, what is the proposed mechanism by which YTHDF2 might affect the interaction between dynein and dynactin? This is not discussed but is central to the model shown in Figure 8.
4. Figure 3C: what distinguishes between binding of misfolded polypeptides and binding mRNAs associated with nascent polypeptides? Did the authors perform these experiments in the presence of RNase?
5. Figure S3B: It appears that each of the components tested are recovered at similar efficiencies to the FLAG-UPF1, in contrast to the authors' claim that hyperphosphorylated UPF1 is more highly associated with YTHDF2 and CED.

Responses to reviewer #1's comments

Summary

This is a nice study where the authors report that in HeLa cells, the N6-methyladenosine (m6A) RNA-binding protein YTHDF2 plays a role in protein quality control. Specifically, they find that YTHDF2 plays a role in aggresome formation through its interaction with UPF1 and the misfolded proteins targeted for degradation. UPF1 is required for YTHDF2's interaction with the CED complex (CTIF, eEF1a1, and DCTN1). YTHDF2 is required to increase the interaction between dynein and the misfolded polypeptide-associated CED complex and UPF1 needed for directed movement to the centrosome and formation of the aggresome. Downregulation of YTHDF2 leads to no aggresome and increased apoptosis. YTHDF2's RNA binding activity was not required for these effects, and this function was not due to the intrinsically disordered domain, but specifically to its interaction with UPF1.

General

The authors addressed the question of how YTHDF2 plays a role in protein turnover in HeLa cells through well-designed experiments, and many different types of methodologies. The data are clear, and visually pleasing, and largely support the claims of the authors. The authors further support their findings with multiple different approaches including both endogenously labeled misfolded proteins, as well as overexpressed proteins known to aggregate. It was also well-written, and easy to read and follow. There are quite a lot of figures, but they are large and clear, and everything is legible, which is much appreciated. They used very detailed approaches to answering their questions with overexpression, knockdown, mutations, and depletions. My biggest weakness is cell type and relevance. Because there have been recent papers showing that use of the aggresome is cell-type specific, how well these findings would apply to work in more relevant cell types is unclear. How much other pathways to aggresome formation (HDAC6, BAG3, etc.) apply in these cells is also unclear, and may want to be added in the discussion. The ease of use in working with the heavily mutated HeLa cells does allow for many experiments to be done that might not be possible in primary cell

types. Perhaps the authors may want to make some type of discussion about aggresome formation and cancer, since their work suggests that in a cancer line (HeLa cells), lack of aggresome formation leads to apoptosis, showing that it would be protective. If there were any other relationship they could show to support their findings in other cancer cells or lines would also be helpful and provide relevance. I have many comments about the manuscript below, however, I recommend to accept with minor revisions required.

- **We appreciate the reviewer's interest in our work and the valuable comments. The reviewer's comments on cell type specificity and other aggresome-targeting cellular machineries were described in the Discussion section as follows: “In this study, we also observed that YTHDF2 increases the circularity of aggresomes of SOD1(G93A), SYN1, and GPx1-Ter (Fig. 6), all of which form spherical aggresomes. These results suggest a possible role of YTHDF2 in the spatiotemporal distribution or condensation of misfolded polypeptides within the aggresome as well as its role in aggresomal targeting of misfolded polypeptides. In particular, given that (i) some aggresome-targeted polypeptides, such as presenilin and GFAP, form ribbon-shaped aggresomes instead of spherical aggresomes^{61, 62} and (ii) a mutant variant of ATP7B forms either spherical or a ribbon-shaped aggresomes depending on cell types⁶³, YTHDF2 may function as a discriminating factor that determines the shape of aggresomes, depending on its substrates and cell types. Furthermore, although we propose a role of YTHDF2 in aggresome formation via CED-UPF1 complex in this study, two other known aggresome-targeting cellular machineries, HDAC6 and BAG3, may interplay with YTHDF2. Indeed, a previous study suggested that CED complex associates with HDAC6 and BAG3 and acts in concert with them for efficient aggresome formation³². Therefore, it is most likely that YTHDF2 may contribute comparably to the circularity of aggresomes formed by an action of HDAC6 and BAG3.”**
- **The comments related to a possible relationship between YTHDF2-mediated aggresome formation and cancers have also been described in the Discussion section of the revised manuscript as follows: “Most current studies on YTHDF2 have focused on the role of YTHDF2 as a reader protein for m⁶A**

RNAs. However, on the basis of our present study, we would like to emphasize that many biological and physiological events mediated by m⁶A (or YTHDF2) should be more carefully evaluated in terms of the two-fold roles of YTHDF2 as an m⁶A reader and aggresome-targeting factor. Furthermore, considering that (i) YTHDF2 plays a protective role in cell survival by promoting aggresome formation (Fig. 5), (ii) aggresomes predict poor outcomes in the pathogenesis of several cancers^{64, 65}, and (iii) YTHDF2 is very closely associated with cancer progression^{66, 67}, a role of YTHDF2 as an aggresome-targeting factor as well as m⁶A reader needs more comprehensive investigation in the future”.

Major

- None

Minor

- I have difficulty accepting the word “dispersed.” Typically dispersed suggests that things were once together but now are no longer so, such as “the crowd dispersed.” Because most of the data are fixed endpoints, it is not clear if the “dispersed” phenotype occurred because things didn’t get to the aggresome, or that they were at the aggresome and then it fell apart (which is what most will intuit with the word “dispersed”). Unless specific data are done such as a timelapse supporting the latter situation, the word should be changed.

- **We agree with the reviewer's suggestion. The word "dispersed" has been removed throughout the manuscript. Instead, we have modified the sentences appropriately depending on the context. For instance, the original sentence “Notably, aggresomes containing polypeptidyl-puro were significantly dispersed upon the downregulation of YTHDF2 or DCTN1 (a component of the CED complex) using small interfering RNA (siRNA) (Fig. 1a,b)” was changed to “Notably, the percentage of cells exhibiting aggresomes was**

significantly reduced upon the downregulation of YTHDF2 or DCTN1 (a component of the CED complex) using small interfering RNA (siRNA) (Fig. 1a,b)."

- Please add that the cells you are working with are HeLa cells in the abstract and early in the Results section as it was not clear until much later.

➤ **As suggested, we have added a description of HeLa cells to the Abstract and early in the Results section.**

- In the introduction, the authors say that "aggresomes" are similar to inclusion bodies, which is contrary to specific studies showing that specific cargoes go to inclusion bodies versus aggresomes; aggresomes are soluble compartments (JUNQ), whereas inclusion bodies are insoluble (iPOD) (Kaganovich et al, 2008 Nature; Ogradnik et al, 2014 PNAS) Please modify.

➤ **We agree with the reviewer's comment regarding the potential confusion caused by the use of similar terminology. To enhance clarity, we have revised the sentence as follows: "aggresomes containing misfolded polypeptides are biochemically and morphologically similar to Lewy bodies (spherical cytoplasmic inclusions containing α -synuclein aggregates) observed in Parkinson's disease and Lewy body dementia⁸⁻¹⁰, thus highlighting the physiological importance of aggresomes."**

➤ **In addition, we would like to point out that aggresomes are different from JUNQ, as aggresomes are insoluble compartments, unlike JUNQ, as described in the cited paper.**

- Reference 15, which is used throughout the introduction for citation of important statements about aggresomes is a preprint written by an author at a community college

who has not studied the aggresome (per Pubmed). This is not acceptable for supporting such critical statements. Use a different citation.

➤ **As recommended, we have removed the original reference in the revised manuscript and instead cited other relevant references.**

- Knockdown of METTL3 and METTL14 was not very complete per western blot. Can you confirm that no RNAs in those conditions are m6A-modified? Statement in the text says "...in contrast, downregulation of the m6A writer...marginally affected aggresome formation." There is barely a reduction of METTL3 and 14 following knockdown and it was not quantified. The authors cannot make that claim unless they accurately describe in the text that that percentage knockdown did not affect aggresome formation. Alternatively, better knockdown could be attempted. As is, there's not much that can be claimed when there is so little knockdown. Further, METTL14 appears to also be reduced with METTL3 knockdown. Again, quantification of the knockdown should be reported.

➤ **As pointed out by the reviewer, we quantitated the band intensities in all western blots obtained from three biological replicates, and the normalized band intensities are presented in the revised Extended Data Fig. 1c. The quantitation results revealed that METTL3, METTL14, and YTHDF2 were downregulated by 40%, 30%, and 40% of normal, respectively..**

➤ **We consistently observed that METTL3 siRNA specifically downregulated METTL3 without affecting the METTL14 protein level. In contrast, METTL14 siRNA downregulated both METTL3 and METTL14 proteins. These findings suggest a possible role of METTL14 in the protein stability of METTL3. In fact, a previous study demonstrated that the association between METTL14 and METTL3 affects the stability of each protein (Wang et al., 2014, Nature Cell Biology).**

- For all of the graphs such as in Figure 1b, Fig2 b,d, etc., authors should report what the DMSO treated condition looked like with all of these knockdowns to suggest if the phenotype is specific to show.

- **The aim of these experiments was to investigate a positive role of m⁶A-related proteins in aggresome formation. For this purpose, we needed to establish cellular conditions favorable for aggresome formation. That was why we always treated cells with MG132. In contrast, under DMSO-treated conditions, aggresomes were barely detectable, and transfection with siRNA against the tested proteins did not trigger any aggresome formation. Due to a lack of additional information, we did not include counting data for DMSO-treated conditions in the original manuscript.**
- **However, to ease the reviewer's concern, the relevant experiments conducted under DMSO-treated conditions are provided as a Figure for the reviewer only. As mentioned earlier, we were not able to observe any aggresomal structures under DMSO-treated conditions.**

Figure for the reviewer only. Aggresome structure is not detectable under DMSO-treated conditions. As performed in Fig. 1a, except that cells were treated with DMSO.

- Fig 1 legend says all data represent at least 2 biological replicates, n=2. These are HeLa cells. If they were primary cells that were hard to obtain and work with, maybe that might be OK, but there should be minimally an n of 3 for this work.

➤ **As commented, we repeated the same experiment during revision. The data obtained from three independently performed experiments are shown in Fig. 1b in the revised manuscript.**

- Fig S4B should be put into the main figure. This finding is important.

➤ **This comment is also related to another comment of the reviewer on the original Fig. 4c (see below). As suggested, the original Extended Data Fig. 4b was moved to the main Fig. 4a, and the original Fig. 4c was moved to Extended Data Fig. 4b in the revised manuscript.**

- The movies, while very cool, don't suggest microtubule movements. The movies are short, with no directionality, is there movement towards the centrosome? This looks like a fence and picket type movement as is.

➤ **We appreciate the reviewer's comment for bringing to our attention the need to investigate the movement of misfolded polypeptides towards the aggresome along the microtubule. In response, we conducted new experiments and added the data as Extended Data Fig 6d-f and Supplementary Video 2. Our data show that misfolded polypeptides move towards the microtubule organizing center, which is the typical location for aggresome formation and detection, along the microtubules. These data provide evidence that misfolded polypeptides exhibit microtubule-based movements towards the aggresome.**

- Statement in line 356 suggests that there is direct evidence that YTHDF2 plays a role in recruiting aggregates to the CED-motor protein complex, leading to the formation of

the aggresome. What you have shown in this paper is interaction but no directionality of recruitment or movement towards the aggresome.

- **As mentioned in your comment, we acknowledge that there is currently no direct evidence for our statement regarding the function of YTHDF2 in recruiting aggregates to the CED-complex. Due to technical limitations, it was not feasible for us to simultaneously track the movements of each misfolded polypeptide, YTHDF2, and the CED-motor protein complex by imaging. However, our new data presented in Extended Data Fig. 6d-f demonstrates the processivity of misfolded polypeptides with microtubules, and we have previously shown the interaction between misfolded polypeptides and dynein in the presence of YTHDF2. Based on these findings, we hypothesize that YTHDF2 may facilitate the processivity of misfolded polypeptides by maintaining their interactions with dynein. Nonetheless, since there is no direct evidence for YTHDF2's role in recruiting aggregates to the CED-complex, we have revised our statement to read as follows: “Taken together, our observations suggest that YTHDF2 plays an important role in the transport of aggregates along microtubules, which may facilitate the formation of the aggresome.”**

- Please put the information in lines 866-868 about total lysate and what those gradients stand for in the western blots in every figure legend with that serial dilution (for e.g. Fig 1)

- **As suggested, the description was added to each Figure legend.**

- Put Fig 4C into the supplement

- **This comment is also related to another comment of the reviewer on the original Extended Data Fig. 4b (see above). As suggested, the original Extended Data Fig. 4b put into the main Fig. 4a. And the original Fig. 4c was moved to Extended Data Fig. 4b in the revised manuscript.**

- In the image in Fig 8C, the location of YTHDF2 doesn't really represent what the authors suggested for its requirement in the co-IP where dynein was detected (Fig 8). How better might this be represented to show that relationship?

➤ **The model was properly revised to represent what we found in this study.**

Responses to reviewer #2's comments

The manuscript of Hwang et al investigates the role of YTHDF2 in aggresome formation. Overall, the work is interesting but some explanations and clarifications are needed before the manuscript can be considered for publication.

➤ **We appreciate the reviewer's interest in our work and the valuable comments.**

Major points:

1) Figures 1, 2 and 4. It is not clear to me what the authors are evaluating here, which makes it difficult to judge the rest of the work. They are comparing distinct aggresomes versus dispersed or lack of aggresomes per cell. What is the definition of a "dispersed aggresome"? From a visual inspection of the images, I do not reconstruct the bar graphs (1b, 2b, 2d and 4b) in these figures. The authors need to clearly explain to the readers how they exactly perform this evaluation (is it the brightness, size, number, of the puncta that indicate aggresomes? or what information is used) so that the reader can follow the paper.

➤ **We apologize for the confusing description. By definition, aggresomes are cytoplasmic inclusions of misfolded polypeptides located at a minus-end of microtubule (centrosome). Using confocal microscopy, one or two inclusions in cells (or puncta) are easily detected upon MG132 treatment. Throughout the manuscript, we counted the number of cells either containing or not containing aggresomes of misfolded polypeptides among the cells that expressed the tested misfolded polypeptides, and presented the percentage of cells either containing or lacking aggresomes. Therefore, all relevant Figures and Figure legends were revised as follows: "Percentage of cells either containing or lacking aggresomes."**

2) Page 13: " The super-resolved structure of the aggresome was not spherically shaped, implying that the aggresome is not in liquid phase". The authors should provide

appropriate controls to demonstrate that the non-spherical shapes are not an artifact of the fixation procedure.

- **Thank you for your comment. We took the opportunity to confirm our super-resolution imaging data by comparing circularity results obtained from live and fixed HeLa cells. We found no significant differences between the two datasets, whether from control or YTHDF2 knockdown conditions.**

- **This was also confirmed with the stress granule, which are well-defined liquid droplets, labeling G3BP1 with Dendra2. We found no significant differences in their shapes even after the fixation protocol. Based on these observations, we conclude that aggregate shapes are not affected by the fixation procedure.**

- **However, the physical properties of the aggregate itself were not of great interest in this paper, so we deleted that sentence from the text and did not include the above data.**

3) Page 15 and Figure 7. There are many inconsistencies with the single particle tracking analysis that should be clarified.

a) Line 333. "We noted that the α value for SYN1 was below zero," First of all, I do not see an α value of less than zero in Figure 7a. Do the authors mean "below 1"? If α is less than zero, than can be do to either an incorrect fit, or too many data points used in the MSD calculation (as a rule of thumb, one does not calculate shifts in the MSD larger than 1/4 of the trajectory length, and sometimes one needs to use even shorter shifts for the MSD).

➤ **Thank you very much for your thorough review and feedback. We apologize for the typo in our manuscript. The “zero” should be changed to “one.” However, we have carefully re-analyzed our data and have found that the statement regarding SYN1 is no longer valid. Please refer to our response to point (c) for further details.**

b) Line 335. "resulting in freely diffusive SYN1 aggregates in our measured population." If we are referring to the MSD of the SYN1 in Figure 7a, then one has subdiffusion (i.e. $\alpha < 1$ or anomalous diffusion) and not free diffusion.

➤ **Thank you again for your faithful review on our work. We would like to inform you that after re-analyzing our data, the statement regarding SYN1 is also no longer valid. Please refer to our response to point (c) for further details.**

c) Figure 7a. It would be useful to show the MSD fits in the plots to get an idea of how descriptive the parameters are. Also, an α value of 1.6 in the right most panel does not make sense when considering the entire plot. The MDS looks more linear, implying an $\alpha = 1$. Only the early times (first five data points) show directed motion. Also, in panel 7b, the units are incorrect. The diffusion coefficients should be given in " $\mu\text{m}^2/\text{s}^\alpha$ ".

➤ **First of all, we sincerely appreciate your comments on the α value. As you pointed out, we noticed that the shape of the original graphs seemed to differ from the α values we obtained. We carefully checked all our analysis codes**

related to diffusion analysis and found a mistake. We found that the code was missing the step of taking the logarithm when calculating the alpha value from the MSD-t data in the process of anomalous diffusion analysis. We reanalyzed all the raw data of particle trajectories for all conditions using a modified code, which is now updated in our GitHub. As a result, the α values for all misfolding-prone polypeptides before and after YTHDF2 knock-down were below 1 (sub-diffusion), and there was no significant difference between them. We believe this is because the cellular environment is a confined space, and the aggregates are bound to bulky complexes and move together. Thus, our conclusion is that YTHDF2 affects the speed and displacement of aggregate movement, but does not affect the anomaly. We would like to express our respect to your excellent intuition that helped us finding the mistake in our analysis code.

- **In addition, although we analyzed anomalous diffusion equation, the diffusion coefficient unit is practically expressed as " $\mu\text{m}^2/\text{s}$ ". Our Fig. 7a graphs represent the average values of MSD for numerous individual trajectories for each time interval. Therefore, when we calculated the diffusion coefficient and alpha values, we did not fit the Fig. 7a graphs but instead fitted each single trajectory to obtain the values. Therefore, the alpha values of the diffusion coefficient data points in Fig. 7b are all different for each data point. The distributions in Fig. 7b are intended to show the differences in diffusion coefficient distribution.**

d) Lines 337-339. "It should be noted that, upon depletion of YTHDF2, both the anomalous coefficients and diffusion constants for the tested misfolded polypeptides reduced drastically (Fig. 7a,b)". Actually, this statement is not exactly precise. Indeed, the exponent decreases indicating stronger subdiffusional or anomalous behavior. However, the diffusion coefficient for SOD and SYN1 are pretty much unchanged (when looking at the diffusive part of the MSD plot within the first ~ 1 s). It is true that there is a drastic reduction in the diffusion coefficient of the Halo-FLAG-GPx1-Ter.

➤ **Once again, we appreciate your comments on our MSD analysis. As we mentioned in our response to point (c), our new analysis results confirmed that there is no change in the α values due to YTHDF2. We have described the α value and diffusion coefficient separately in the main text. Thanks to your careful comments on our analysis, we were able to correct our mistakes and rewrite the main text of the entire section.**

e) Figure 7d. The co-diffusion is confusing. In all cases, the spots are separated by ~ 1 μm . If the particles are moving together, I would expect them to be closer together. The authors should explain what they are observing here in more detail.

➤ **This difference is considered to be due to optical aberration. We used dual-viewer equipment to measure both signals simultaneously. We applied approximate colocalization mapping to the phase contrast images of the cell membrane from different channels prior to image merging. Alternatively, there could be a bulky protein complex between YTHDF2 and the aggregates. Despite this, we strongly believe that the observed paired particles we showed in the figures are physically associated because the particles move together consistently throughout the entire track. This cannot happen with two particles in random motion. We explained this in the main text.**

4) Lines 353-355. "We successfully captured the directional motion of YTHDF2, accompanied by aggregates, for all the misfolded peptides we tested (Fig. 7d and Supplementary Video S1d)." The authors need to substantiate this statement better. There is no evidence in Figure 7d and Supplementary Video S1d that the motion is directional nor is the evidence for co-diffusion very strong.

➤ **We agree with the reviewer. We observed that by labeling the microtubule, the aggregates migrate toward the microtubule organizing center, where aggresomes are usually formed. Although the trajectories we showed in**

Extended Data Fig. 6 are not unidirectionally biased 1D motion, it shows that it tends towards the aggresome in general.

Minor points:

1. Line 141-143: " Immunostaining revealed that either FLAG-YTHDF2 or endogenous YTHDF2 was evenly distributed throughout the cytoplasm regardless of MG132 treatment (Fig. 1c and Extended Data Fig. 1d)." I have difficulties understanding the "either" in this sentence. That would imply that sometimes FLAG-YTHDF2 is evenly distributed, and sometimes YTHDF2, but never both at the same time, which does not make sense to me. Please reword the sentence for readability.

➤ **The sentence was revised as follows: “Immunostaining revealed that FLAG-YTHDF2 and endogenous YTHDF2 were evenly distributed throughout the cytoplasm”**

2. Overall, the English is fair in the paper. However, there are a few places where the English detracts from the readability of the text.

➤ **The revised manuscript was proofread by a native speaker to improve readability.**

Responses to reviewer #3's comments

This manuscript from the Kim laboratory investigates potential connections between the m6A reader protein YTHDF2, the RNA decay factor UPF1, and protein quality control. The authors propose that YTHDF2, uniquely among YTHDF paralogs, is important for formation and morphology of aggresomes formed under a high burden of misfolded proteins. The authors provide evidence that depletion of YTHDF2 impairs the coalescence of dispersed protein aggregates into aggresomes, which they link to increased sensitivity to proteotoxic stress. Loss of YTHDF2 is also associated with an apparent reduction in the interaction between the CED complex and dynein, which in turn impairs transport of aggregates to the aggresome. Based on co-IP and knockdown experiments, the authors develop a model where UPF1 recruits YTHDF2 to the CED complex. The mechanism by which YTHDF2 association with the CED complex promotes aggresome formation is undefined. The possibility of links between RNA and protein quality control pathways is intriguing, but I have many reservations about the manuscript as currently presented. The significance and regulatory rationale of the involvement of YTHDF2 in aggresome formation is unclear, and little mechanistic insight is provided. I also have concerns about data quality and interpretation. Please see below for detailed comments.

➤ **We appreciate the reviewer's valuable comments.**

Major points:

1. Figure 1 and S1C: The authors use METTL3 and METTL14 knockdown to ask whether the effect of YTHDF2 on aggresome formation is dependent on m6A. However, S1C shows that substantial amounts of both m6A writer proteins persist in the knockdown conditions. Given the presence of METTL3/14 in the assayed cells, it is inappropriate to use these data to conclude that the role of YTHDF2 is m6A-independent.

➤ **As pointed out, the band intensities in all western blots obtained from three biological replicates were quantified, and the normalized band intensities are**

presented in the revised Extended Data Fig. 1c. The quantification results showed that METTL3, METTL14, and YTHDF2 were downregulated by 40%, 30%, and 40% of normal, respectively. Although the data presented in Fig. 1 may not completely support the m⁶A-independent role of YTHDF2 in aggresome formation due to insufficient downregulation, subsequent complementation experiments using YTHDF2 variants lacking m⁶A-binding ability clearly demonstrate that YTHDF2 functions in aggresome formation in an m⁶A-independent manner (Fig. 4).

2. In Figure 1, 2, and 4, the authors use IF to examine aggresome formation, and these are key pieces of data for their argument. However, it is not clear to me how aggresome formation was quantified in these experiments. The Y axes of the graphs in 1B, 2B, 2D, 4B say "Relative distribution of [polypeptidyl puro or CFTR-deltaF508]" but the figure legends say that this reports the "Relative percentages of cells containing either aggresomes or dispersed aggregates." I don't think these descriptions are interchangeable (the axis labels imply quantification of signal inside and outside of aggresomes, which would be preferable). I also don't understand what is meant by "relative percentage" in these figure legends. Most importantly, I could not find a description of the criteria used to distinguish an aggresome from dispersed aggregates. This is critical for interpretation of the data.

- **We apologize for the confusing description. By definition, aggresomes are cytoplasmic inclusions of misfolded polypeptides located at a minus-end of microtubule (centrosome). Using confocal microscopy, one or two inclusions in cells (or puncta) are easily detected upon MG132 treatment. Throughout the manuscript, we counted the number of cells either containing or not containing aggresomes of misfolded polypeptides among the cells that expressed the tested misfolded polypeptides, and presented the numbers as a percentage. Therefore, all relevant Figures and Figure legends were revised as follows: "Percentage of cells either containing or lacking aggresomes."**

3. Figure 2A, the overall signal in YTHDF3 cells appears lower. What accounts for this and how might it affect the quantification?

- **As described in our response to Reviewer's Comment #2, we counted the number of cells containing or lacking aggregates of misfolded polypeptides and presented the percentages in Fig. 2b. Therefore, the difference in signal intensity does not affect our quantification and our main conclusion.**
- **The reason why the overall signal in YTHDF3 cells appears lower is simply due to a lower cell density in culture plate. We captured images incidentally in a region with a low cell density. Of note, we did not observe any significant growth defect of cells transfected with YTHDF3 siRNA. For clarity, the original Fig. 2a was replaced by a revised Fig. 2a including more representative images for YTHDF3.**

4. Figure S2B: It is not possible to exclude that YTHDF2 knockdown interferes with microtubule organization based on these images in the absence of quantification.

- **We agree with the reviewer. If YTHDF2 downregulation significantly affects microtubule polymerization as observed in cells treated with nocodazole, we would definitely have quantitated the data. However, as you can see the confocal data, we were not able to observe any significant changes in microtubule polymerization. Because of this reason, we would like to keep the original confocal images without further quantitation.**

5. Throughout the quantification of the westerns is questionable. Virtually every standard deviation is +/- 0.0 or 0.1, which is unlikely if the experimental error is normally distributed.

- **We presented all quantitative data without any alterations. All raw data were also uploaded as source data. Probably, the reason why the reviewer think like that is because of the rounding of the second digit after the decimal point.**

6. Figure 4A: Does the expression of wild-type YTHDF2 faithfully rescue aggresome morphology in this assay? These aggresomes don't look like those in the control and the overall signal is faint. If this is a representative image, I have doubts about the validity of the knockdown-rescue system used. As above, the question of how aggresomes vs aggregates were quantified is essential.

- **During revision, the order of panels was changed. The Fig. 4a,b in the original manuscript has been moved to Fig. 4b,c, as suggested by reviewer #1. In the revised Fig. 4b, the expression of YTHDF2-WT almost completely restored the aggresome, as also presented in quantitation results in Fig. 4c. The reason why the reviewer pointed out may be due to the difference in intensities between the control cells and YTHDF2-WT expressing cells. Therefore, we replaced the original images with new representative images in the revised manuscript.**
- **Please also note that, as mentioned earlier in our response to comment #2 of this reviewer, we quantified the number of cells containing or lacking aggresomes of misfolded polypeptides among the cells expressing the tested misfolded polypeptides, and presented the results as a percentage.**

7. Figure 7: I don't understand the basis for this statement, given the data shown: "Collectively, our observations from the single-particle tracking analysis indicate that YTHDF2 enhances the processivity of misfolded polypeptide movements and the average moving distance of small aggregates per association with the microtubule without detaching." Did the authors perform analyses of misfolded polypeptide processivity in the presence and absence of YTHDF2? I don't think that the aggregate measures of displacement shown in figure 7 give direct insight into processivity.

- **We appreciate the reviewer's comment for bringing to our attention the need to investigate the movement of misfolded polypeptides towards the aggresome along the microtubule. In response, we conducted new experiments and added the data as Extended Data Fig 6d-f. Our data show that misfolded**

polypeptides move towards the microtubule organizing center, which is the typical location for aggresome formation and detection, along the microtubules. These data provide evidence that misfolded polypeptides exhibit microtubule-based movements towards the aggresome. Although the trajectories we showed in Extended Data Fig. 6 are not unidirectionally biased 1D motion, they do show a general trend towards the aggresome.

8. Figure 7: It's not clear to me why YTHDF2 would have an effect on SYN1 displacement that is similar to the other two model misfolded proteins, since the authors seem to be suggesting that the amount of free SYN1 protein limits the ability to assay directed motion/assisted diffusion: "We noted that the α value for SYN1 was below zero, which may be due to excessive overexpression of the protein compared to capable motor proteins or CED complex, resulting in freely diffusive SYN1 aggregates in our measured population." Can the authors explain further?

- **We apologize for confusing the reviewer. During the revision process, we discovered that there was a mistake in our diffusion analysis codes. In the results of reanalyzing the raw data, it was confirmed that all misfolding prone polypeptides showed a similar tendency in alpha values. For more details, please refer to our response to reviewer 2's comment 3-c.**

Minor points:

1. How do the differences in aggresome shape fit into the model? A discussion of this would be helpful, since that phenotype was highlighted in the abstract.

- **As commented, a brief description on the YTHDF2-mediated increase in the circularity of aggresome has been added to the Discussion section as follows: "In this study, we also observed that YTHDF2 increases the circularity of aggresomes of SOD1(G93A), SYN1, and GPx1-Ter (Fig. 6), all of which form spherical aggresomes. These results suggest a possible role of YTHDF2 in the spatiotemporal distribution or condensation of misfolded polypeptides within**

the aggresome as well as its role in aggresomal targeting of misfolded polypeptides. In particular, given that (i) some aggresome-targeted polypeptides, such as presenilin and GFAP, form ribbon-shaped aggresomes instead of spherical aggresomes^{61, 62} and (ii) a mutant variant of ATP7B forms either spherical or a ribbon-shaped aggresomes depending on cell types⁶³, YTHDF2 may function as a discriminating factor that determines the shape of aggresomes, depending on its substrates and cell types. Furthermore, although we propose a role of YTHDF2 in aggresome formation via CED-UPF1 complex in this study, two other known aggresome-targeting cellular machineries, HDAC6 and BAG3, may interplay with YTHDF2. Indeed, a previous study suggested that CED complex associates with HDAC6 and BAG3 and acts in concert with them for efficient aggresome formation³². Therefore, it is most likely that YTHDF2 may contribute comparably to the circularity of aggresomes formed by an action of HDAC6 and BAG3”.

2. Figure 1C: The significance of finding FLAG-YTHDF2-N in aggresomes following MG132 treatment isn't clear to me. "These data suggest that the N-terminal half of YTHDF2 has molecular features associated with protein misfolding or aggresome formation." Is a very vague statement. This could be said of any misfolded protein and does not give insight into the function of YTHDF2.

- **For a clearer description, the sentence was revised as follows: “the N-terminal half of YTHDF2 has intrinsic features that can induce protein misfolding and aggresome formation.”**

3. Also, what is the proposed mechanism by which YTHDF2 might affect the interaction between dynein and dynactin? This is not discussed but is central to the model shown in Figure 8.

- **We currently lack a precise understanding of the molecular mechanism underlying this event. As a result, we have added a brief explanation of this phenomenon to the figure legend in the revised manuscript.**

4. Figure 3C: what distinguishes between binding of misfolded polypeptides and binding mRNAs associated with nascent polypeptides? Did the authors perform these experiments in the presence of RNase?

- **Thank you for your insightful comments. As suggested, we repeated the same IP experiments using cell extracts that were either treated or not treated with RNase A during the revision process. Although RNase A treatment significantly reduced the levels of endogenous transcripts, comparable reductions in the levels of co-immunoprecipitated polypeptidyl-puro were observed in both RNase A-treated and untreated samples following UPF1 downregulation. These results suggest that RNA is not significantly involved in the association between YTHDF2 and misfolded polypeptides. As a result, we replaced the original Fig. 3c with the revised Fig. 3c and added the related validation experiment to Extended Data Fig. 3c in the revised manuscript.**

5. Figure S3B: It appears that each of the components tested are recovered at similar efficiencies to the FLAG-UPF1, in contrast to the authors' claim that hyperphosphorylated UPF1 is more highly associated with YTHDF2 and CED.

- **It appears that there may be some misunderstanding. In our data, YTHDF2 copurified with FLAG-UPF1-HP by 2.2-fold more than FLAG-UPF1-WT. In addition, CTIF copurified with FLAG-UPF1-HP by 2.5-fold more than FLAG-UPF1-WT. Therefore, all of the IP data presented in Extended Fig. 3b support our claim that “a greater amount of YTHDF2 and CED components coimmunoprecipitated with UPF1-HP relative to UPF1-WT and UPF1-HP-12A.”**

Reviewers' Comments:

Reviewer #1:

Remarks to the Author:

The authors addressed almost all of my concerns. However, I would like to point out 2 specific items:

1. "However, to ease the reviewer's concern, the relevant experiments conducted under DMSO-treated conditions are provided as a Figure for the reviewer only. As mentioned earlier, we were not able to observe any aggresomal structures under DMSO-treated conditions."

I think this should be added to the Supplementary figures for the general article audience as well, not just "for the reviewer only."

2. "In addition, we would like to point out that aggresomes are different from JUNQ, as aggresomes are insoluble compartments, unlike JUNQ, as described in the cited paper."

In response, I would say that based on the literature, what the cargo is in the aggresome is likely what drives the solubility. For example, in Ogrodnik et al, 2014 the JUNQ is surrounded by a vimentin cage, and this structure is transitory. This vimentin cage is a generalized feature and within the general definition of aggresomes (Johnston et al, 1998). Additionally, in Morrow et al, 2020, in neural stem cells, the polyubiquitin-rich aggresome at the centrosome was also surrounded by a vimentin cage. This structure was also transitory and easily cleared in stem cells, and was visualized in the soluble fractions of westerns. Garcia-Mata et al, 1999 demonstrated that aggresomes can be formed by soluble, nonubiquitinated proteins. In Johnston et al, 1998, Ron Kopito's initial paper describing the aggresome using CFTR, they found that CFTR was initially soluble, then changed with time into insoluble. Thus, multiple factors, likely cell type, cell state, and what proteins are enriched in the aggresome - whether endogenous polyubiquitinated proteins, or overexpressed mutated proteins - likely drive differences in the solubility of the aggresome.

The changes the authors have made in their text about inclusion bodies are acceptable.

I suggest the revisions be accepted, and the paper published.

Reviewer #2:

Remarks to the Author:

The revised manuscript is improved and the authors have adequately dealt with most of the criticisms. There are a couple minor points that could still be improved:

Figure 6d: The number of significant digits on the probability should be corrected to 7.3×10^{-5} .

Figure 7a: Reduce the number of digits for alpha to only the significant digits (most likely 2). What do the lines represent? If they are only there to guide the eye, leave them out.

Figure 7b: When using the formula $MSD = 4Dt^\alpha$ for fitting the MSD, the correct units for D are $\mu\text{m}^2/\text{s}^\alpha$ (and alpha should be included unless it is very close to 1). The labels on the y-axes should be corrected.

Reviewer #3:

Remarks to the Author:

The revised manuscript is improved in some respects, but several key questions remain. Several of the claims made by the authors are not adequately supported by the data shown. For specific comments, please see below.

1. The authors' response to my original point 1 (insufficiency of relatively poor knockdown of METTL3 and METTL14 to claim an m6A-independent function of YTHDF2), acknowledges the shortcoming of this line of argument, but the manuscript still uses these data to claim an m6A-independent role. It is true that the later m6A binding-deficient mutants are consistent with that interpretation, but the experiments shown in Figure 1 and S1C remain inconclusive with respect to the claim of m6A independence in that section of the paper.

2. I am confused by the authors' response to my original point 4 (validity of the conclusion that YTHDF2 knockdown does not affect microtubule organization). The text says that "YTHDF2 downregulation only marginally affected global microtubule formation and distribution," while the response says that they were "not able to observe any significant changes." In the absence of quantification, how can the authors make these claims? What do they mean by "only marginally affected" and "significant changes" if those statements are not based on some sort of quantitative analysis? This is an important point, as the entire interpretation of the YTHDF2 knockdown results are dependent on a lack of cytoskeletal alteration.

3. In reference to my original point 7 and referee #2's point 4, the data still do not substantiate the claim that YTHDF2 "enhances the processivity of misfolded polypeptide movements" or show motion that appears consistent with directional motion of dynein on microtubules. The added Figure S6 shows only a small number of single-particle traces, which again are difficult to interpret lacking quantitative analysis.

Responses to reviewer #1's comments

The authors addressed almost all of my concerns. However, I would like to point out 2 specific items:

➤ **We appreciate the reviewer's valuable comments.**

1. "However, to ease the reviewer's concern, the relevant experiments conducted under DMSO-treated conditions are provided as a Figure for the reviewer only. As mentioned earlier, we were not able to observe any aggresomal structures under DMSO-treated conditions."

I think this should be added to the Supplementary figures for the general article audience as well, not just "for the reviewer only."

➤ **As suggested, the figure was added to Extended Data Fig. 1a in the revised manuscript.**

2. "In addition, we would like to point out that aggresomes are different from JUNQ, as aggresomes are insoluble compartments, unlike JUNQ, as described in the cited paper."

In response, I would say that based on the literature, what the cargo is in the aggresome is likely what drives the solubility. For example, in Ogradnik et al, 2014 the JUNQ is surrounded by a vimentin cage, and this structure is transitory. This vimentin cage is a generalized feature and within the general definition of aggresomes (Johnston et al, 1998). Additionally, in Morrow et al, 2020, in neural stem cells, the polyubiquitin-rich aggresome at the centrosome was also surrounded by a vimentin cage. This structure was also transitory and easily cleared in stem cells, and was visualized in the soluble fractions of westerns. Garcia-Mata et al, 1999 demonstrated that aggresomes can be formed by soluble, nonubiquitinated proteins. In Johnston et al, 1998, Ron Kopito's initial paper describing the aggresome using CFTR, they found that CFTR was initially soluble, then changed with time into insoluble. Thus, multiple factors, likely cell type,

cell state, and what proteins are enriched in the aggresome - whether endogenous polyubiquitinated proteins, or overexpressed mutated proteins - likely drive differences in the solubility of the aggresome.

The changes the authors have made in their text about inclusion bodies are acceptable.

- **We are grateful for the reviewer's clear and insightful explanation of the concept of aggresome.**

I suggest the revisions be accepted, and the paper published.

- **We appreciate the reviewer's time and efforts in evaluating our manuscript.**

Responses to reviewer #2's comments

The revised manuscript is improved and the authors have adequately dealt with most of the criticisms. There are a couple minor points that could still be improved:

- **We appreciate the reviewer's valuable comments.**

Figure 6d: The number of significant digits on the probability should be corrected to 7.3×10^{-5} .

- **Thank you for your comment on our work. As you recommended, for Fig. 6d, we revised the number of significant digits to 7.3×10^{-5} .**

Figure 7a: Reduce the number of digits for alpha to only the significant digits (most likely 2). What do the lines represent? If they are only there to guide the eye, leave them out.

- **As suggested, we reduced the number of digits for alpha values to two significant digits. In addition, for your question on the lines in our graphs, lines are simply connecting the datapoints on our MSD plots. As you pointed out in your comment, they function as a guidance for the audience to follow the datapoints and simply deliver the message that there is an increase in MSD values over time. Thus, we removed the lines in our graphs as you suggested.**

Figure 7b: When using the formula $MSD = 4Dt^\alpha$ for fitting the MSD, the correct units for D are $\mu\text{m}^2/\text{s}^\alpha$ (and alpha should be included unless it is very close to 1). The labels on the y-axes should be corrected.

- **We appreciate your comment on our work. As you pointed out, we corrected the unit for D values to $\mu\text{m}^2/\text{s}^\alpha$. To help the audience for better understanding, we also included alpha values in our source data.**

Responses to reviewer #3's comments

The revised manuscript is improved in some respects, but several key questions remain. Several of the claims made by the authors are not adequately supported by the data shown. For specific comments, please see below.

- **We appreciate the reviewer's valuable comments.**

1. The authors' response to my original point 1 (insufficiency of relatively poor knockdown of METTL3 and METTL14 to claim an m6A-independent function of YTHDF2), acknowledges the shortcoming of this line of argument, but the manuscript still uses these data to claim an m6A-independent role. It is true that the later m6A binding-deficient mutants are consistent with that interpretation, but the experiments shown in Figure 1 and S1C remain inconclusive with respect to the claim of m6A independence in that section of the paper.

- **We concur with the reviewer's suggestion. Accordingly, we have now commenced the description of the m⁶A-independent role of YTHDF2 in aggresome formation from Fig. 4. Prior to this, we have only speculated about the possibility as follows: "These results suggest that YTHDF2 promotes the efficient formation of aggresomes of misfolded polypeptides, possibly in an m⁶A-independent manner".**

2. I am confused by the authors' response to my original point 4 (validity of the conclusion that YTHDF2 knockdown does not affect microtubule organization). The text says that "YTHDF2 downregulation only marginally affected global microtubule formation and distribution," while the response says that they were "not able to observe any significant changes." In the absence of quantification, how can the authors make these claims? What do they mean by "only marginally affected" and "significant changes" if those statements are not based on some sort of quantitative analysis? This is an important point, as the entire interpretation of the YTHDF2 knockdown results are dependent on a lack of cytoskeletal alteration.

- **To address the reviewer's critical comments, we set up the experiments that enables us to determine the amount of microtubule content versus free-tubulin content in a cell population. The new results revealed that a treatment with nocodazole almost completely disrupted microtubules. In contrast, downregulation of either UPF1 or YTHDF2 did not significantly affect microtubule formation. These new data have been added to Extended Data Fig. 2c in the revised manuscript.**

3. In reference to my original point 7 and referee #2's point 4, the data still do not substantiate the claim that YTHDF2 "enhances the processivity of misfolded polypeptide movements" or show motion that appears consistent with directional motion of dynein on microtubules. The added Figure S6 shows only a small number of single-particle traces, which again are difficult to interpret lacking quantitative analysis.

- **Thank you for your kind comments on our work. As you mentioned in your remark, we agree that in Extended Data Fig. 6 only a small number of single particles were tracked. We tried to repeat this experiment many times in order to increase the number of samples and generate quantitatively better results. However, due to technical limitations, it was very challenging to find the tracks on the microtubules. One possible reason is that each protein may move in 3-dimensional space. If the particle moves below or above the focal plane of our microscopic setting, the final analyzed track may seem like simple diffusive movement. Moreover, as we mentioned in our manuscript, we could only observe small portion of target proteins on the microtubules since they were sparsely labeled. Therefore, now we corrected our claims throughout the manuscript as follows: “facilitating the diffusion dynamics of the movement of misfolded” (Abstract), “YTHDF2 enhances the diffusion dynamics of misfolded polypeptides during movement” (page 14), “consequently facilitating the movement of dynein motor proteins for the efficient transport of small protein aggregates toward the aggresome” (page 17), “promotes movement of the complex” (page 17), “promotes the dynein-mediated retrograde transport” (page 44), and “Example traces of misfolded polypeptides associated with the microtubule network in the presence of YTHDF2” (page 47).**